# A LAW OF ADVERSARIAL RISK, INTERPOLATION, AND LABEL NOISE

**Daniel Paleka** *
ETH Zurich
daniel.paleka@inf.ethz.ch

**Amartya Sanyal** *
ETH AI Center, ETH Zurich
amartya.sanyal@ai.ethz.ch

## ABSTRACT

In supervised learning, it has been shown that label noise in the data can be interpolated without penalties on test accuracy. We show that interpolating label noise induces adversarial vulnerability, and prove the first theorem showing the relationship between label noise and adversarial risk for any data distribution. Our results are almost tight if we do not make any assumptions on the inductive bias of the learning algorithm. We then investigate how different components of this problem affect this result including properties of the distribution. We also discuss non-uniform label noise distributions; and prove a new theorem showing uniform label noise induces nearly as large an adversarial risk as the worst poisoning with the same noise rate. Then, we provide theoretical and empirical evidence that uniform label noise is more harmful than typical real-world label noise. Finally, we show how inductive biases amplify the effect of label noise and argue the need for future work in this direction.

## 1 INTRODUCTION

Label noise is ubiquitous in data collected from the real world. Such noise can be a result of both malicious intent as well as human error. The well-known work of Zhang et al. (2017) observes that training overparameterised neural networks with gradient descent can memorize large amounts of label noise without increased test error. Recently, Bartlett et al. (2020) investigated this phenomenon and termed it *benign overfitting*: perfect interpolation of the noisy training dataset still leads to satisfactory generalization for overparameterized models. A long series of works (Donhauser et al., 2022; Hastie et al., 2022; Muthukumar et al., 2020) focused on providing generalization guarantees for models that interpolate data under uniform label noise. This suggests that noisy training data does not hurt the test error of overparameterized models.

However, when deploying machine learning systems in the real world, it is not enough to guarantee low test error. Adversarial vulnerability is a practical security threat (Kurakin et al., 2016; Sharif et al., 2016; Eykholt et al., 2018) for deploying machine learning algorithms in critical environments. An adversarially vulnerable classifier, that is accurate on the test distribution, can be forced to err on carefully perturbed inputs even when the perturbations are small. This has motivated a large body of work towards improving the *adversarial robustness* of neural networks (Goodfellow et al., 2014; Papernot et al., 2016; Tramèr et al., 2018; Sanyal et al., 2018; Cisse et al., 2017). Despite the empirical advances, the theoretical guarantees on robust defenses are still poorly understood.

Consider the setting of uniformly random label noise. Under certain distributional assumptions, Sanyal et al. (2021) claim that with moderate amount of label noise, when training classifiers to zero training error, the adversarial risk is always large, even when the test error is low. Experimentally, this is supported by Zhu et al. (2021), who showed that common methods for reducing adversarial risk like adversarial training in fact does not memorise label noise. However, it is not clear whether their distributional assumptions are realistic, or if their result is tight. To deploy machine learning models responsibly, it is important to understand the extent to which a common phenomenon like label noise can negatively impact adversarial robustness. In this work, we improve upon previous theoretical results, proving a stronger result on how label noise *guarantees* adversarial risk for large enough sample size.

---

*Equal contribution.

On the other hand, existing experimental results (Sanyal et al., 2021) seem to suggest that neural networks suffer from large adversarial risk even in the small sample size regime. Our results show that this phenomenon cannot be explained without further assumptions on the data distributions, the learning algorithm, or the machine learning model. While specific biases of machine learning models and algorithms (referred to as inductive bias) have usually played a "positive" role in machine learning literature (Vaswani et al., 2017; van Merriënboer et al., 2017; Mingard et al., 2020), we show how some biases can make the model more vulnerable to adversarial risks under noisy interpolation.

Apart from the data distribution and the inductive biases, we also investigate the role of the label noise model. Uniform label noise, also known as random classification noise (Angluin and Laird, 1988), is a natural choice for modeling label noise, but it is neither the most realistic nor the most adversarial noise model. Yet, our results show that when it comes to guaranteeing a lower bound on adversarial risk for interpolating models, uniform label noise model is not much weaker than the optimal poisoning adversary. Our experiments indicate that natural label noise (Wei et al., 2022) is not as bad for adversarial robustness as uniform label noise. Finally, we also attempt to understand the conditions under which such benign (natural) label noise arises.

**Overview**     First, we introduce notation necessary to understand the rest of the paper. Then, we prove a theoretical result (Theorem 2) on adversarial risk caused by label noise, significantly improving upon previous results (Theorem 1 from Sanyal et al. (2021)). In fact, our Theorem 2 gives the first theoretical guarantee that adversarial risk is large for all compactly supported input distributions and all interpolating classifiers, in the presence of label noise. Our theorem does not rely on the particular function class or the training method. Then, in Section 3, we show Theorem 2 is tight without further assumptions, but does not accurately reflect empirical observations on standard datasets. Our hypothesis is that the experimentally shown effect of label noise depends on properties of the distribution and the inductive bias of the function class. In Section 4, we prove (Theorem 5) that uniform label noise is on the same order of harmfmul as worst case data poisoning, given a slight increase in dataset size and adversarial radius. We also run experiments in Figure 3, showing that mistakes done by human labelers are more benign than the same rate of uniform noise. Finally, in Section 5, we show that the inductive bias of the function class makes the impact of label noise on adversarial vulnerability much stronger and provide an example in Theorem 7.

## 2 GUARANTEEING ADVERSARIAL RISK FOR NOISY INTERPOLATORS

**Our setting**     Choose a norm $\|\cdot\|$ on $\mathbb{R}^d$, for example $\|\cdot\|_2$ or $\|\cdot\|_\infty$. For $\boldsymbol{x} \in \mathbb{R}^d$, let $B_r(\boldsymbol{x})$ denote the $\|\cdot\|$-ball of radius $r$ around $\boldsymbol{x}$. Let $\mu$ be a distribution on $\mathbb{R}^d$ and let $f^* : \mathcal{C} \to \{0, 1\}$ be a measurable ground truth classifier. Then we can define the adversarial risk of any classifier $f$ with respect to $f^*, \mu$, given an adversary with perturbation budget $\rho > 0$ under the norm $\|\cdot\|$, as

$$\mathcal{R}_{\mathrm{Adv},\rho}(f, \mu) = \mathbb{P}_{\boldsymbol{x} \sim \mu}\left[\exists \boldsymbol{z} \in \mathcal{B}_\rho(\boldsymbol{x}), \ f^*(\boldsymbol{x}) \neq f(\boldsymbol{z})\right]. \tag{1}$$

Next, consider a training set $((\boldsymbol{z}_1, y_1), \ldots, (\boldsymbol{z}_m, y_m))$ in $\mathbb{R}^d \times \{0, 1\}$, where the $\boldsymbol{z}_i$ are independently sampled from $\mu$, and each $y_i$ equals $f^*(\boldsymbol{z}_i)$ with probability $1 - \eta$, where $\eta > 0$ is the label noise rate. Let $f$ be any classifier which correctly interpolates the training set. We now state the main theoretical result of Sanyal et al. (2021) so that we can compare our result with it.

**Theorem 1** ( Sanyal et al. (2021))**.** *Suppose that there exist $c_1 \geq c_2 > 0$, $\rho > 0$, and a finite set $\zeta \subset \mathbb{R}^d$ satisfying*

$$\mu\left(\bigcup_{\boldsymbol{s} \in \zeta} B_{\rho/2}(\boldsymbol{s})\right) \geq c_1 \quad and \quad \forall \boldsymbol{s} \in \zeta, \ \mu\left(B_{\rho/2}(\boldsymbol{s})\right) \geq \frac{c_2}{|\zeta|} \tag{2}$$

*Further, suppose that each of these balls contains points from a single class. Then for $\delta > 0$, when the number of samples $m \geq \frac{|\zeta|}{\eta c_2} \log\left(\frac{|\zeta|}{\delta}\right)$, with probability $1 - \delta$*

$$\mathcal{R}_{\mathrm{Adv},\rho}(f, \mu) \geq c_1. \tag{3}$$

This is the first guarantee for adversarial risk caused by label noise in the literature. However, Theorem 1 has two extremely strong assumptions:

- The input distribution has mass $c_1$ in a union of balls, each of which has mass at least $c_2$;
- Each ball only contains points from a single class.

It is not clear why such balls would exist for real-world datasets, or even MNIST or CIFAR-10. In Appendix F, we give some evidence against the second assumption in particular. In Theorem 2, we remove these assumptions and show that our guarantees hold for all compactly supported input distributions, with comparable guarantees on adversarial risk.

Let $\mathcal{C}$ be a compact subset of $\mathbb{R}^d$. An important quantity in our theorem will be the *covering number* $N = N(\rho/2; \mathcal{C}, \|\cdot\|)$ of $\mathcal{C}$ in the metric $\|\cdot\|$. The covering number $N$ is the minimum number of $\|\cdot\|$-balls of radius $\rho/2$ such that their union contains $\mathcal{C}$. For any distribution $\mu$ on $\mathbb{R}^d$, denote by $\mu(\mathcal{C}) = \mathbb{P}_{x \sim \mu}[x \in \mathcal{C}]$ the mass of the distribution $\mu$ contained in the compact $\mathcal{C}$.

**Theorem 2.** *Let $\mathcal{C} \subset \mathbb{R}^d$ satisfy $\mu(\mathcal{C}) > 0$, and let $N = N(\rho/2; \mathcal{C}, \|\cdot\|)$ be its covering number. For $\delta > 0$, when the number of samples satisfies $m \geq \frac{8N}{\mu(\mathcal{C})\eta} \log \frac{2N}{\delta}$. with probability $1 - \delta$ we have that*

$$\mathcal{R}_{\mathrm{Adv},\rho}(f, \mu) \geq \frac{1}{4}\mu(\mathcal{C}) \tag{4}$$

*for any classifier $f$ that interpolates the training set.*

The compact $\mathcal{C}$ can be chosen freely, allowing us to make tradeoffs between the required number of samples $m$ and the lower bound on the adversarial risk. As the chosen $\mathcal{C}$ expands in volume, the lower bound on the adversarial risk $\mu(\mathcal{C})$ also increases. However, this also increases the required number of samples for the theorem to kick in, which depends on its covering number $N$. The tradeoff curve depends on the distribution $\mu$; we discuss this in Section 3.

Note that Theorem 2 is easier to interpret than Theorem 1, as it holds for any compact $\mathcal{C}$ as opposed to a finite set $\zeta$ of dense balls. Our result avoids the unwieldy assumptions, and in fact gives a slightly stronger guarantee than Theorem 1. When Equation (2) holds, note that we can choose the compact $\mathcal{C} = \bigcup_{s \in \zeta} B_{\rho/2}(s)$ from Theorem 1 yielding $N = |\zeta|$ and $\mu(\mathcal{C}) = c_1$. Thus, under similar settings as the previous result, our theorem requires the number of samples $m = \widetilde{\Omega}\left(\frac{|\zeta|}{\eta c_1}\right)$, which is smaller than $m = \widetilde{\Omega}\left(\frac{|\zeta|}{\eta c_2}\right)$ required in Theorem 1.

We leave the proof of Theorem 2 to Appendix A, but we provide a brief sketch of the ideas.

*Proof sketch* We want to prove that a large portion of points from $\mu$ have a mislabeled point nearby when $m$ is large enough. The expected number of label noise training points is $\eta m$; however a priori those could be anywhere in the support of $\mu$.

The key idea is that we can always find a set of $\|\cdot\|$-balls covering a lot of measure, with each of the balls having a large enough density of $\mu$. We prove this in Lemma 8 and provide an illustration in Figure 1. The blue dotted circles in Figure 1 are the $\|\cdot\|$-balls; they do not cover the entire space but cover a significant portion of the entire density. Then, if we take a lot of $\|\cdot\|$-balls with large density of a single class, we can prove that label noise induces an opposite-labeled point in each of the chosen balls given $m$ large enough.

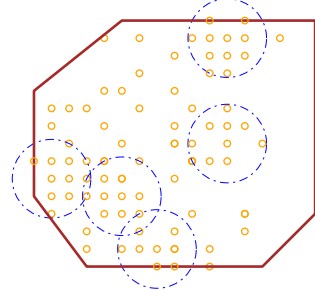

Figure 1: Depending on the covering number of $\mathcal{C}$, a small number of $\|\cdot\|$-balls of sufficient density cover a lot of the measure of $\mathcal{C}$. Label noise makes every point drawn from the covered set adversarially vulnerable.

Concretely, the probability for a single chosen ball to not be adversarially vulnerable is on the order of $\left(1 - \frac{\eta}{2N}\right)^{\mu(\mathcal{C})m}$, and summing this up over the $O(N)$ chosen balls goes to zero when $m$ is large. By the union bound, each of these balls is then adversarially vulnerable, summing up to a constant adversarial risk.

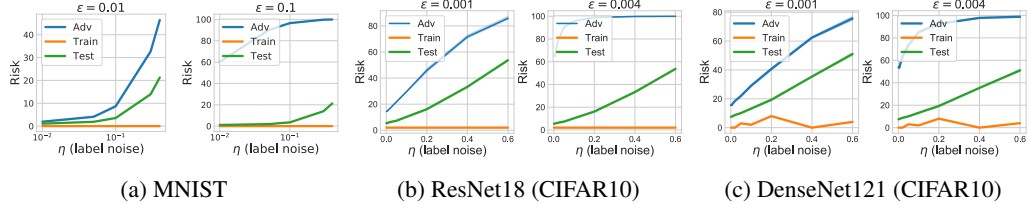

(a) MNIST      (b) ResNet18 (CIFAR10)      (c) DenseNet121 (CIFAR10)

Figure 2: From Sanyal et al. (2021). Adversarial error increases with increasing label noise $\eta$ (x-axis) at a rate much faster than predicted by Theorem 2. Here, $\epsilon$ is the perturbation magnitude ($\rho$ in the current paper). The label noise is synthetically injected in the training set with probability $\eta$.

We considered the binary classification case for simplicity. The proof in Appendix A lower bounds the adversarial risk on a single true class. Thus, by summing up the risks for each class, we lose only a constant factor on the guaranteed adversarial risk in the multi-class case.

For compactly supported $\mu$, we can take $\mathcal{C}$ to be the support of $\mu$ to prove a general statement.

**Corollary 3.** *Let $N$ be the covering number of* $\mathrm{supp}(\mu)$ *with balls of radius $\rho/2$. For $\delta > 0$, when the number of samples satisfies $m \geq \frac{8N}{\eta} \log \frac{2N}{\delta}$. with probability $1 - \delta$ we have that*

$$\mathcal{R}_{\mathrm{Adv},\rho}(f, \mu) \geq \frac{1}{4}. \tag{5}$$

This is easier to understand than Theorem 2: if interpolating a dataset with label noise, the number of samples required to guarantee constant adversarial risk scales with the covering number of the support of the distribution.

*Remark* 1. The proof in Appendix A actually proves Equation (4) by first proving a stronger fact: if $\mu|_{\mathcal{C}}$ is the normalized restricton of $\mu$ on $\mathcal{C}$, then

$$\mathcal{R}_{\mathrm{Adv},\rho}(f, \mu|_{\mathcal{C}}) \geq \frac{1}{4}. \tag{6}$$

We can use this to give better guarantees when $f^*$ is not robust. If a region of $\mathrm{supp}(\mu)$ is already adversarially vulnerable using the true classifier $f^*$, we can omit it from $\mathcal{C}$, and just add the guarantee from Theorem 2 to the original adversarial risk to get a stronger lower bound on $\mathcal{R}_{\mathrm{Adv},\rho}(f, \mu)$.

## 3   PRACTICAL IMPLICATIONS ON SAMPLE SIZE

In this section, we discuss the limitations of results like Theorem 2. When we allow arbitrary interpolating classifiers, we show that Theorem 2 paints an accurate picture of the interaction of label noise, interpolation, and adversarial risk. However, this particular theoretical framework cannot explain the strong effect of label noise on adversarial risk in practice (see Figure 2). We argue that this requires a better understanding of the inductive biases of the hypothesis class and the optimization algorithm.

**Required sample size for Theorem 2**   The number of required samples $m$ in Theorem 2 can be very large, depending on the density and the covering number of the chosen compact $\mathcal{C}$. Consider $\|\cdot\|$ to be the $\|\cdot\|_\infty$ norm, as is customary in adversarial robustness research (Goodfellow et al., 2014). Then the balls $B_\rho$ are small hypercubes in $\mathbb{R}^d$. If we choose $\mathcal{C}$ to be the hypercube $[0, 1]^d$, the covering number scales exponentially in dimension:

$$N = N(\rho; [0, 1]^d, \|\cdot\|_\infty) \simeq \left(\frac{1}{\rho}\right)^d. \tag{7}$$

A rough back-of-the-envelope calculation indicates that this can scale badly even for standard datasets such as MNIST ($d = 784$) or CIFAR-10 ($d = 3072$), since in Theorem 2 we need $m \gtrsim \frac{N}{\mu(\mathcal{C})\eta}$. This amounts an impossibly large sample size ($m \gtrsim 10^{784}$) for $\rho = 0.2$ to explain the effect already observed with in $m = 50000$ MNIST training samples in Figure 2a.

Hence our result often does not guarantee any adversarial risk if the number of samples $m$ is small. In general, the covering number of a dataset is not polynomial in the dimension, except if the data has special properties in the given metric. For example, if the data distribution is supported on a subspace of $\mathbb{R}^d$ of ambient dimension $k < d$, we can pick a $\mathcal{C}$ for which the covering number in $\|\cdot\|_2$ will depend only on $k$ and not on $d$. However, this is still not sufficient to explain the behaviour in Figures 2b and 2c. If our result indeed kicks in for some ambient dimension $k$ with DenseNet121 on CIFAR10, then for a given adversarial risk (say $80\%$), the power law dependence would imply $\left(\frac{\rho_2}{\rho_1}\right)^k \approx \frac{\eta_1}{\eta_2}$, where label noise rate $\eta_i$ yields adversarial error $80\%$ with perturbation budget $\rho_i$. With another back-of-the-envelope calculation using Figure 2c, we set $\eta_1, \eta_2 = 0.7, 0.1$ and $\rho_1, \rho_2 = 0.001, 0.004$. This yields an ambient dimension $k < 2$, which is unrealistic for CIFAR-10.

These calculations suggest the possibility that a tighter bound than Theorem 2 might exist. However, the large sample size is not just a limitation of Theorem 2. In fact, we show that if arbitrary classifiers and distributions are allowed, the adversarial risk cannot be lower bounded for $m = \text{poly}(d)$.

**Our result is tight** It is a priori possible that the true dependence of adversarial risk on label noise kicks in for much lower sample size regimes than in Theorem 2. This might suggest that the lower bound on sample complexity can be improved. We show this is not the case and in fact our bound is sharp. In particular, we design a simple distribution on $\mathbb{R}^d$ such that there exist classifiers which correctly and robustly interpolate datasets with the number of samples $m$ exponential in $d$.

**Proposition 4.** *Let $\mu$ be the uniform distribution on $\mathbb{S}^{d-1} = \left\{x_1, \ldots, x_d \in \mathbb{R}^d : x_1^2 + \ldots + x_d^2 = 1\right\}$, and let the ground truth classifier $f^*$ be a threshold function on $x_1$: $f^*(x) = \mathbb{1}_{x_1 > \frac{1}{2}}$. Consider any adversarial radius $\rho < \frac{1}{4}$ in the Euclidean metric. Then, for any label noise $\eta < 1$: with high probability, there exists a classifier $f$ that interpolates $m = \lfloor 1.01^d \rfloor$ samples from the label noise distribution, such that $\mathcal{R}_{\text{Adv}, \rho}(f, \mu) = o_d(1)$.*

*Proof sketch* The main ingredient of the proof is the concentration of measure on $\mathbb{S}^{d-1}$, which makes the training samples far apart in the Euclidean metric. We leave the full proof to Appendix B. Similar statements in the clean data setting have appeared before, e.g. in Bubeck and Sellke (2021).

Note that Proposition 4 shows a construction where Theorem 2 cannot guarantee an adversarial risk lower bound with sample size $m$ sub-exponential in $d$. Hence, the covering number of any substantial portion of $\mathbb{S}^{d-1}$ is exponential in the dimension $d$. This unintentionally proves the well known fact that the covering number of the sphere $\mathbb{S}^{d-1}$ in the Euclidean metric is exponential.[1]

**Optimizing $\mathcal{C}$ can avoid large sample size** While Proposition 4 shows the tightness of Theorem 2 in the worst case, it is possible a smaller sample size requirement is sufficient under certain conditions. In particular, if we can pick a compact $\mathcal{C}$ with small covering number, such that the measure $\mu(\mathcal{C})$ is large, then Theorem 2 allows for a small sample size while guaranteeing a large adversarial risk.

*Example* Take an adversarial radius $\rho > 0$ in the $\|\cdot\|_\infty$ metric and choose $r \in (0, \frac{1}{2})$, let $\mu = (1 - r)\mu_1 + r\mu_2$ be the average of two measures, $\mu_1$ and $\mu_2$, with $\mu_1$ the uniform distribution on $[0, 1]^d$, and $\mu_2$ the uniform distribution on a smaller hypercube $[0, \rho]^d$.

The first choice $\mathcal{C} = [0, 1]^d$ as in Corollary 3 has covering number on the order of $\rho^{-d}$. Theorem 2 is then vacuous until $m \gtrsim \rho^{-d}/\eta$, which is very large in high dimensions. Note that this is necessary to achieve the lower bound of adversarial risk of $\frac{1}{4}$. However, if we only want to guarantee an adversarial risk of $\frac{r}{4}$, instead we can use $\mathcal{C} = [0, \rho]^d$. For this, the covering number is 1 and we can use Theorem 2 for $m = O\left(\frac{1}{\eta}\right)$. This suggests that while the required sample size for the maximal adversarial risk is possibly very large, it can be much smaller, depending on the distribution, for guaranteeing a smaller adversarial risk.

---

[1] See Proposition 4.16 in `https://www.stats.ox.ac.uk/~rebeschi/teaching/AFoL/20/material/lecture04.pdf`

Formally, to get the "best possible" $m$ in Theorem 2 for a certain adversarial risk lower bound $r$, we should solve the following optimization problem over subsets of $\text{supp}(\mu)$:

$$\min_{\mu(\mathcal{C}) \geq 4r} \frac{N(\rho/2, \mathcal{C}, \|\cdot\|_\infty) \log N(\rho/2, \mathcal{C}, \|\cdot\|_\infty)}{\mu(\mathcal{C})}. \tag{8}$$

The above optimization problem comes from substituting $r$ into the adversarial risk placeholder in Theorem 2. Equation (8) provides a complexity measure to get tighter lower bounds on the adversarial vulnerability induced by uniform label noise. It is not known whether the optimization is tractable in general. However, the concept of having to solve an optimization problem in order to get a tight lower bound is common in the literature. Some examples are the *representation dimension* (Beimel et al., 2019) and the *SQ dimension* (Feldman, 2017) .

To conclude, this section shows that in real world data, the required sample size for guaranteeing large adversarial risk from interpolating label noise is significantly smaller than what an off-the-shelf application of Theorem 2 might suggest. However, we also proved that it is not possible to obtain tighter bounds without further assumptions on the data or the model.

## 4 NON-UNIFORM LABEL NOISE

In previous sections, we discussed guaranteeing a lower bound on adversarial error for noisy interpolators in Section 2. In Section 3, we discussed the tightness of the said bound. However, all of these results assumed that the label noise is distributed uniformly on the points in the training set, which corresponds to the popular Random Classification Noise (Angluin and Laird, 1988) model. However, an uniform noise model is not very realistic (Hedderich et al., 2021; Wei et al., 2022); and it is thus sensible to also investigate how our results change under non-uniform label noise models.

**Uniform noise is almost as harmful as poisoning** The worst-case non-uniform label noise model is *data poisoning*, where an adversary can choose the labels of a subset of the training set of a fixed size (see Biggio and Roli (2018) for a survey). It is well known that flipping the label of a constant number of points in the training set can significantly increase the error of a logistic regression model (Jagielski et al., 2018) or an SVM classifier (Biggio et al., 2011). On the contrary, the test error of a neural networks has been surprisingly difficult to hurt by data poisoning attacks which flip a small fraction of the labels. Lu et al. (2022) show that, on some datasets, the effect of adversarial data poisoning on test accuracy is, in fact, comparable to the effect of uniform label noise.

We phrase the main result of this section informally in Theorem 5, using standard game-theoretic metaphors for data poisoning, and defer the formal version to Appendix C. Let again $\mu$ be a distribution on $\mathbb{R}^d$ and $f^*$ a correct binary classifier, and let $\eta$ be the label noise rate. Consider a game in which an adversary flips the labels of a subset of the training set, and tries to maximize the minimum adversarial risk among all interpolators of the noisy (after flipping labels) training set. We will compare the performances of two adversaries:

- **Uniform**, who samples $T$ points uniformly from the distribution, and flips the label of each of the $T$ points in the sampled training set with probability $\eta$;
- **Poisoner**, who inserts $N = \eta m$ arbitrary points from $\text{supp}(\mu)$ with flipped labels into the training set and then samples the remaining $m - N$ points uniformly, with correct labels.

Here $T$ and $m$ are the respective training set sizes. If $T \sim \frac{1}{\eta} N = m$, then the two adversaries flip the same number of labels in expectation. In that sense, both of these adversaries have the same budget. However, the Poisoner can choose which points to flip and thus intuitively, in this regime, the Poisoner will get a higher adversarial risk than the Uniform. Surprisingly, we can prove the Uniform is not much worse if $T \sim m \log m$.

**Theorem 5** (Informal statement of Theorem 11). *Denote the adversarial risks of the Uniform and the Poisoner adversaries by $\mathcal{R}^{Unif}$ and $\mathcal{R}^{Poison}$ respectively. For any $\rho > 0$, we have that*

$$\mathcal{R}^{Unif}_{2\rho} \geq \frac{1}{2} \mathcal{R}^{Poison}_\rho \tag{9}$$

*as long as $\mathcal{R}^{Poison}_\rho = \Omega(1)$ and $T \gtrsim m \log m$.*

Roughly speaking, the above theorem shows that if the Uniform adversary is given *double the adversarial radius* and *a log factor increase on the training set size*, then Uniform can guarantee an adversarial risk of the same magnitude as the Poisoner. The full statement and the proof of the theorem are given in Appendix C but we provide a brief sketch here.

*Proof sketch* The Poisoner will choose $N$ points to flip, adversarially poisoning every point in the $N$ corresponding $\rho$-balls. As in Theorem 2, we can use Lemma 8 to show that a subset of the balls with density $\Omega(1/N)$ covers half of the adversarially vulnerable region. Then Uniform samples $T$ points, and we expect to hit each of the balls in the chosen subset. Because of the doubled radius, each sampled point makes the the whole $\rho$-ball vulnerable. The log factor comes from the same reason as in the standard coupon collector (balls and bins) problem; if we have $N$ bins with hitting probabilities $\Omega(1/N)$, then we need $\Omega(N \log N)$ tries to hit each bin at least once.

**Some label noise models are benign**  Different label noise models with the same expected label noise rate can have very different effects on the adversarial risk. In the previous sections, we showed that uniform label noise is almost as bad as the worst possible noise model with the same label noise rate. This raises the question whether all noise models are as harmful as the uniform label noise model. We answer the question in the negative especially for data distributions that have a *long tailed* structure: many far-apart low-density subpopulations in the support of the distribution $\mu$.

For this, we show a simple data distribution $\mu$ in Proposition 6, where:

- Uniform label noise with probability $\eta$ guarantees adversarial risk on the order of $\eta$;
- A different label noise model, with expected label noise rate $\eta$, which affects only *the long tail* of the distribution $\mu$ can be interpolated with $o(1)$ adversarial risk.

We argue that this is neither an unrealistic distributional assumption nor an impractical noise model. In fact, most standard image datasets, like SUN (Xiao et al., 2010), PASCAL (Everingham et al., 2010), and CelebA (Liu et al., 2015) have a long tail (Zhu et al., 2014; Sanyal et al., 2022). Moreover, it is natural to assume that mistakes in the datasets are more likely to occur in the long tail, where the data points are atypical. In Feldman (2020), it was argued that noisy labels on the long tail are one of the reasons for why overparameterized neural networks remember the training data. Formally, we prove the following regarding the benign noise model for a long-tailed distribution.

**Proposition 6.** *Let $A < B$ be integers with $A$ much smaller than $B$. Let $\mu$ be a mixture model on $\mathbb{R}$ supported on a disjoint union of $A + B$ intervals, such that half of the mass is on the first $A$ intervals and half of the mass is on the last $B$ intervals:*

$$\mu = \frac{1}{2A} \sum_{i=1}^{A} \mathrm{Unif}\left(i, i + \frac{1}{2}\right) + \frac{1}{2B} \sum_{j=1}^{B} \mathrm{Unif}\left(A + j, A + j + \frac{1}{2}\right)$$

*Let the ground truth label be zero everywhere. Sample two datasets $\mathcal{D}_1, \mathcal{D}_2$ of size $m$ from $\mu$ using two different label noise distributions: For $\mathcal{D}_1$, flip the label of each sample $x \in [0, A + B]$ independently with probability $\eta$. For $\mathcal{D}_2$, flip the label of each sample $x \in [A, A + B]$ independently with probability $2\eta$, and leave the labels of the other samples unchanged. Then, for any $\rho, \delta \in \left(0, \frac{1}{2}\right)$, for the number of samples $m = \tilde{\Theta}_\rho(A)$ (ignoring log terms), we have that with probability $1 - \delta$:*

- *For any $f$ which interpolates $\mathcal{D}_1$, the adversarial risk is large: $\mathcal{R}_{\mathrm{Adv},\rho}(f, \mu) = \Omega_\rho(1)$.*

- *There exists $f$ which interpolates $\mathcal{D}_2$, such that $\mathcal{R}_{\mathrm{Adv},\rho}(f, \mu) = O_\rho\left(\frac{A}{B}\right)$.*

A similar distribution was previously used as a representative long tailed distribution in the context of privacy and fairness in Sanyal et al. (2022). Our result can also be extended to more complicated long-tailed distributions with a similar strategy. Proposition 6 implies that the the first noise model (for $\mathcal{D}_1$) induces $\Omega(1)$ adversarial risk on all interpolators. On the other hand, for the second noise model i.e. for $\mathcal{D}_2$, it is possible to obtain interpolators with adversarial risk on the order of $\left(\frac{A}{B}\right)$. Thus, for distributions where $A \ll B$, this implies the existence of almost robust interpolators despite having the same label noise rate.

**Real-world noise is more benign than uniform label noise**  To support our argument that real world noise models are, in fact, more benign than uniform noise models, we consider the noise

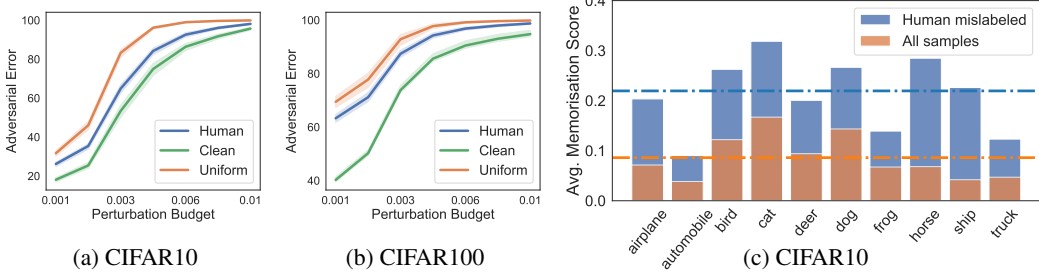

(a) CIFAR10      (b) CIFAR100      (c) CIFAR10

Figure 3: Figures 3a and 3b plots adversarial risk against perturbation budget on CIFAR10 and CIFAR100 datasets respectively, with three label noise models. Clean denotes the original CIFAR-10/100 labels. Human denotes the human-generated labels from (Wei et al., 2022). Uniform refers to uniformly random label noise with the same rate as Human. Figure 3c plots the average memorisation score per class for: Human mislabeled examples i.e. examples with Human noisy labels and All samples. The horizontal line is the average across all classes. The higher score of human mislabeled examples indicates that those examples belong to the long tail of the distribution.

induced by human annotators in Wei et al. (2022). They propose a new version of the CIFAR10/100 dataset (Krizhevsky et al., 2009) where each image is labelled by three human annotators. Known as CIFAR-10/100-n, each example's label is decided by a majority vote on the three annotated labels. We train ResNet34 models till interpolation on these two datasets. The label noise rate, after the majority vote, is $\approx 9\%$ in CIFAR10 and $\approx 40\%$ in CIFAR100. We repeat the same experiment for uniform label noise with the same noise rates, and also without any label noise. Each of these models' adversarial error is evaluated with an $\ell_\infty$ PGD adversary plotted in Figures 3a and 3b.

Figures 3a and 3b show that, for both CIFAR10 and CIFAR100, uniform label noise is indeed worse for adversarial risk than human-generated label noise. For CIFAR-10, the model that interpolates human-generated label noise is almost as robust as the model trained on clean data. This supports our argument that real-world label noise is more benign, for adversarial risk, than uniform label noise.

An important direction for future research is understanding what types of label noise models are useful mathematical proxies for realistic label noise. We shed some light on this question using the idea of *memorisation score* (Feldman and Zhang, 2020). Informally, memorisation score quantifies the atypicality of a sample; it measures the increase in the loss on a data point when the learning algorithm does not observe it during training compared to when it does. A high memorisation score indicates that the point is unique in the training data and thus, likely, lies in the long tail of the distribution. In Figure 3c, we plot the average memorisation score of each class of CIFAR10 in brown, and the average for images that were mislabeled by the human annotator in blue. It is clearly evident that the mislabeled images have a higher memorisation score. This supports our hypothesis (also in Feldman (2020)) that, in the real world, examples in the long tail, are more likely to be mislabeled.

## 5 THE ROLE OF INDUCTIVE BIAS

We have seen, in Section 3, that without further assumptions the theoretical guarantees in Theorem 2 only hold for very large training sets. In this section, we discuss how the inductive bias of the hypothesis class or the learning algorithm can lower the sample size requirement and point to recent work that indicates the presence of such inductive biases in neural networks.

**Inductive bias can hurt robustness even further** There is ample empirical evidence (Ortiz-Jimenez et al., 2020; Kalimeris et al., 2019; Shamir et al., 2021) that neural networks exhibit an inductive bias that is different from what is required for robustness. Shah et al. (2020) also provides empirical evidence that neural networks exhibit a certain inductive bias, that they call simplicity bias, that hurts adversarial robustness. Ortiz-Jiménez et al. (2022) show that this is also responsible for a phenomenon known as catastrophic overfitting.

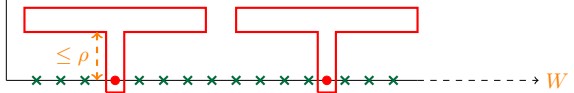

Figure 4: Visualization of a portion of the distribution $\mu$ and the hypothesis class $\mathcal{H}$ used in Theorem 7. The crosses are the mislabeled examples and the circles are correctly labelled examples. All the circles are adversarially vulnerable to upwards perturbations of magnitude less than $\rho$.

Here, we show a simple example to illustrate the role of inductive bias. Consider a binary classification problem on a data distribution $\mu$ and a dataset $S_{m,\eta}$ of $m$ points, sampled i.i.d. from $\mu$ such that the label of each example is flipped with probability $\eta$.

**Theorem 7.** *For any $\rho > 0$, there exists a distribution $\mu$ on $\mathbb{R}^2$ and two hypothesis classes $\mathcal{H}$ and $\mathcal{F}$, such that for any label noise rate $\eta \in (0, 1/2)$ and dataset size $m = \Theta\left(\frac{1}{\eta}\right)$, in expectation we have that: for all $h \in \mathcal{H}$ that interpolate $S_{m,\eta}$,*

$$\mathcal{R}_{\mathrm{Adv},\rho}(h,\mu) \geq \Omega(1); \tag{10}$$

*whereas there exists an $f \in \mathcal{F}$ that interpolates $S_{m,\eta}$ and $\mathcal{R}_{\mathrm{Adv},\rho}(f;\mu) = \mathcal{O}(\rho)$.*

The classes $\mathcal{H}$ and $\mathcal{F}$ are precisely defined in Appendix E where a formal proof is given as well, but we provide a proof sketch here. The data distribution $\mu$ in Theorem 7 is uniform on the set $[0, W] \times \{0\}$, that is, the data is just supported on the first coordinate where $W \gg \rho > 0$. The ground truth is a threshold function on the first coordinate. The constructed $f \in \mathcal{F}$ simply labels everything according to the ground truth classifier (which is a threshold function on the first coordinate) except the mislabeled data points; where it constructs infinitesimally small intervals around the point on the first coordinate. Note that this construction is similar to the one in Proposition 4. By design, it interpolates the training set and its expected adversarial risk is upper bounded by $2m\eta\rho$.

Each hypothesis in the hypothesis class $\mathcal{H}$ can be thought of as a union of T-shaped decision regions. The region inside the T-shaped regions are classified as $1$ and the rest as $0$. Note that the "head" of a T-shaped region make the region on the data manifold (first coordinate) directly below them adversarially vulnerable. The width of the T can be interpreted as the inductive bias of the learning algorithm. The decision boundaries of neural networks usually lie on the data manifold (Somepalli et al., 2022); and the network behaves more smoothly off the data manifold. A natural consequence of this is that the head of the Ts are large. This is not the exact explanation of inductive bias in neural networks, but rather an illustrative example for what might be happening in practice. In Appendix G, we provide experimental evidence to show this type of behaviour for neural networks.

There are two important properties of this simple example relevant for understanding adversarial vulnerability of neural networks. First, the adversarial examples constructed here are off-manifold: they do not lie on the manifold of the data. This has been observed in prior works (Hendrycks and Gimpel, 2016; Sanyal et al., 2018; Khoury and Hadfield-Menell, 2018). Second, our examples implicitly exhibit the *dimpled manifold* phenomenon recently described in Shamir et al. (2021).

**Is Theorem 2 about the wrong function class?** When fitting deep neural networks to real datasets, the results of Theorem 2 still hold even when the number of samples $m$ is much smaller than required, as can be seen in Figure 2. We think that proving guarantees on adversarial risk in the presence of label noise is within reach for simple neural network settings. Towards this goal, we propose a conjecture in a similar vein to Bubeck et al. (2020):

**Conjecture 1.** Let $f : \mathbb{R}^d \to \mathbb{R}$ be a neural network with a single hidden layer with $k$ neurons. Under the same conditions as in Theorem 2, for the number of samples $m = \widetilde{\Omega}(\frac{1}{\eta}\mathrm{poly}(k, d))$,

$$\mathcal{R}_{\mathrm{Adv},\rho}(f,\mu) \geq \mathrm{const}. \tag{11}$$

for a distribution $\mu$ supported on $[0,1]^d$.

In short, we conjecture that neural networks exhibit inductive biases which hurt robustness when interpolating label noise. Dohmatob (2021) show a similar result for a large class of neural networks. However, the Bayes optimal error in their setting is a positive constant (as opposed to zero in our setting), and we assume uniform label noise in the training set. Understanding these properties is important for training on real-world data, where label noise is not a possibility but rather a norm.

ACKNOWLEDGMENTS

Amartya Sanyal acknowledges the ETH AI Center for the postdoctoral fellowship. We thank Mislav Balunović for discussing and checking the proof of Theorem 2, Domagoj Bradač for discussing Lemma 8, and Jacob Clarysse and Florian Tramèr for general feedback.

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

## A    PROOF OF THEOREM 2

Here we prove the following statement:

**Theorem 2.** *Let $\mathcal{C} \subset \mathbb{R}^d$ satisfy $\mu(\mathcal{C}) > 0$, and let $N = N(\rho/2; \mathcal{C}, \|\cdot\|)$ be its covering number. For $\delta > 0$, when the number of samples satisfies $m \geq \frac{8N}{\mu(\mathcal{C})\eta} \log \frac{2N}{\delta}$. with probability $1 - \delta$ we have that*

$$\mathcal{R}_{\mathrm{Adv},\rho}(f, \mu) \geq \frac{1}{4}\mu(\mathcal{C}) \tag{4}$$

*for any classifier $f$ that interpolates the training set.*

For notational convenience, we replace $\rho$ by $2\rho$ in all places for the proof below.

*Proof.* Without loss of generality, let $\mathcal{C}_0 = \{\, \boldsymbol{x} \in \mathcal{C} : f^*(\boldsymbol{x}) = 0 \,\}$ have probability $\mu(\mathcal{C}_0) \geq \frac{1}{2}\mu(\mathcal{C})$. Let $\mu_0 = \mu|_{\mathcal{C}_0}$, normalized so that $\mu_0(\mathcal{C}_0) = 1$.

By Chernoff, with probability $1 - \exp\left(-\frac{\mu(\mathcal{C})m}{16}\right) \geq 1 - \frac{\delta}{2}$, at least $m_0 = \lfloor\frac{\mu(\mathcal{C})m}{4}\rfloor$ of the samples $\boldsymbol{z}_i$ are in $\mathcal{C}_0$. Without loss of generality, let $\boldsymbol{z}_1, \ldots, \boldsymbol{z}_{m_0}$ be those samples. Then

$$\mathcal{R}_{\mathrm{Adv},2\rho}(f, \mu) \geq \frac{1}{2}\mu(\mathcal{C})\, \mathbb{P}_{\boldsymbol{x} \sim \mu, \boldsymbol{x} \in \mathcal{C}_0}\left[\exists \boldsymbol{z} \in \mathcal{B}_{2\rho}(\boldsymbol{x}),\ f^*(\boldsymbol{x}) \neq f(\boldsymbol{z}))\right] \tag{12}$$

$$= \frac{1}{2}\mu(\mathcal{C})\, \mathbb{P}_{\boldsymbol{x} \sim \mu_0}\left[\exists \boldsymbol{z} \in \mathcal{B}_{2\rho}(\boldsymbol{x}),\ f(\boldsymbol{z}) \neq 0\right] \tag{13}$$

$$\geq \frac{1}{2}\mu(\mathcal{C})\, \mathbb{P}_{\boldsymbol{x} \sim \mu_0}\left[\exists\, i \leq m_0\ :\ \boldsymbol{z}_i \in \mathcal{B}_{2\rho}(\boldsymbol{x}) \cap \mathcal{C}_0,\ f(\boldsymbol{z}_i) \neq 0\right] \tag{14}$$

$$= \frac{1}{2}\mu(\mathcal{C})\, \mathbb{P}_{\boldsymbol{x} \sim \mu_0}\left[\exists\, i \leq m_0\ :\ \boldsymbol{x} \in \mathcal{B}_{2\rho}(\boldsymbol{z}_i),\ \boldsymbol{z}_i \in \mathcal{C}_0,\ f(\boldsymbol{z}_i) \neq 0\right]. \tag{15}$$

$$= \frac{1}{2}\mu(\mathcal{C})\, \mu_0\left(\bigcup_{i \leq m_0,\ f(\boldsymbol{z}_i) \neq 0} \mathcal{B}_{2\rho}(\boldsymbol{z_i})\right). \tag{16}$$

Let $\boldsymbol{s}_1, \ldots, \boldsymbol{s}_N$ be the centers of a minimum $\rho$-covering of $\mathcal{C}_0$.

The plan is the following: we will lower bound $\bigcup_{i \leq m_0,\ f(\boldsymbol{z}_i) \neq 0} \mathcal{B}_{2\rho}(\boldsymbol{z}_i)$ by the union of some $\mathcal{B}_\rho(\boldsymbol{s}_k)$, which will have large $\mu_0$-measure in total. Moreover, each of the chosen $\mathcal{B}_\rho(\boldsymbol{s}_k)$ will have large enough $\mu_0$-measure. For this, we use the following general lemma:

**Lemma 8.** *Let $\boldsymbol{s}_1, \ldots, \boldsymbol{s}_N$ be the centers of some balls $\mathcal{B}_r(\boldsymbol{s}_i)$ in $\mathbb{R}^d$, and take any measure $\nu$. Then, for any constant $0 < \alpha < 1$, there exists a subset $S \subseteq \{\, 1, \ldots, N \,\}$ of the balls such that:*

- $\nu\left(\bigcup_{i \in S} \mathcal{B}_r(\boldsymbol{s}_i)\right) \geq (1 - \alpha)\, \nu\left(\bigcup_{i=1}^{N} \mathcal{B}_r(\boldsymbol{s}_i)\right).$

- $\nu\left(\mathcal{B}_r(\boldsymbol{s}_i)\right) \geq \frac{\alpha}{N}\, \nu\left(\bigcup_{i=1}^{N} \mathcal{B}_r(\boldsymbol{s}_i)\right)$ *for all $i \in S$.*

Informally, the first condition says that the union of the chosen subset has a constant fraction of the measure of the union. The second condition says that each of the chosen balls has $\Omega(1/N)$ of the measure of the union of all balls.

*Proof.* Without loss of generality, let the balls be ordered by measure:

$$\nu\left(\mathcal{B}_r(\boldsymbol{s}_1)\right) \geq \nu\left(\mathcal{B}_r(\boldsymbol{s}_2)\right) \geq \cdots \geq \nu\left(\mathcal{B}_r(\boldsymbol{s}_N)\right). \tag{17}$$

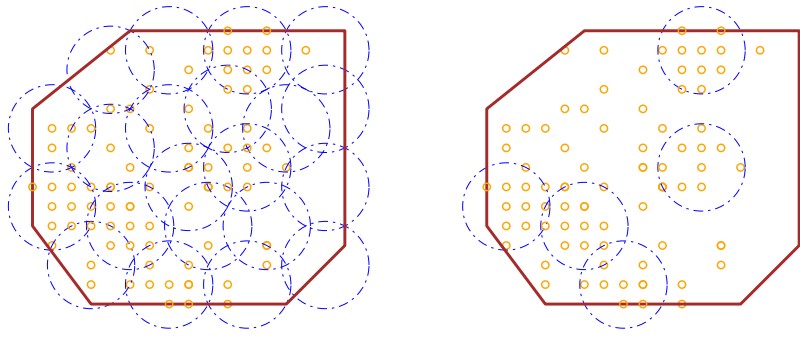

a) The original cover of $\mathcal{C}$.  b) The dense greedy subcover.

Figure 5: Illustration of Lemma 8 and Corollary 9. Given a cover of $N$ balls, we can pick a subcover of balls covering at least half of the measure, with each ball having measure at least $\frac{1}{2N}$.

We take the greedy subset $S = \{1, \ldots, K\}$, where $1 \leq K \leq N$ is the largest index such that $\nu\left(\mathcal{B}_r(s_K)\right) \geq \alpha \, \nu\left(\bigcup_{i=1}^N \mathcal{B}_r(s_i)\right)$.

$$\nu\left(\bigcup_{i=1}^K \mathcal{B}_r(s_i)\right) = \nu\left(\bigcup_{i=1}^N \mathcal{B}_r(s_i)\right) - \nu\left(\bigcup_{i=K+1}^N \mathcal{B}_r(s_i)\right) \tag{18}$$

$$\geq \nu\left(\bigcup_{i=1}^N \mathcal{B}_r(s_i)\right) - (N-K)\frac{\alpha}{N} \, \nu\left(\bigcup_{i=1}^N \mathcal{B}_r(s_i)\right) \tag{19}$$

$$\geq (1-\alpha) \, \nu\left(\bigcup_{i=1}^N \mathcal{B}_r(s_i)\right). \tag{20}$$

The first inequality follows because for all $i \geq K+1$ it holds $\nu\left(\mathcal{B}_r(s_K)\right) < \alpha \, \nu\left(\bigcup_{i=1}^N \mathcal{B}_r(s_i)\right)$, and the second is because $\frac{N-K}{N} \leq 1$. ∎

We can apply the above to the situation in the proof of Theorem 2 with $\alpha = \frac{1}{2}$. Without loss of generality, order the covering $s_1, \ldots, s_N$ by the $\mu_0$-measure of the corresponding balls:

$$\mu_0\left(\mathcal{B}_\rho(s_1)\right) \geq \mu_0\left(\mathcal{B}_\rho(s_2)\right) \geq \cdots \geq \mu_0\left(\mathcal{B}_\rho(s_N)\right). \tag{21}$$

**Corollary 9.** *If $1 \leq K \leq N$ is the largest index such that $\mu_0(\mathcal{B}_\rho(s_K) \geq \frac{1}{2N}$, then*

$$\mu_0\left(\bigcup_{k=1}^K \mathcal{B}_\rho(s_i)\right) > \frac{1}{2}. \tag{22}$$

We now show that the chosen balls are dense enough to get samples in the training set with high probability.

**Lemma 10.** *With probability $1 - \delta/2$, each $\mathcal{B}_\rho(s_k)$ for $k \leq K$ contains at least one $z_i \in \mathcal{C}_0$ such that $f(z_i) \neq 0$.*

*Proof.* We have

$$\mathbb{P}\left[z_i \in \mathcal{B}_\rho(s_k) \mid z_i \in \mathcal{C}_0\right] \geq \frac{1}{2N}, \tag{23}$$

and because the label corruption is independent from everything, we also have

$$\mathbb{P}\left[f(z_i) \neq 0 \mid z_i \in \mathcal{C}_0\right] = \eta \tag{24}$$

$$\implies \mathbb{P}\left[f(z_i) \neq 0 \text{ and } z_i \in \mathcal{B}_\rho(s_k) \mid z_i \in \mathcal{C}_0\right] \geq \frac{\eta}{2N} \tag{25}$$

Therefore,

$$\mathbb{P}\left[\mathcal{B}_\rho(\boldsymbol{s}_k) \cap \{\, \boldsymbol{z}_i : i \le m_0, \ f(\boldsymbol{z}_i) \ne 0 \,\} = \emptyset\right] \tag{26}$$

$$= \prod_{i=1}^{m_0} \mathbb{P}\left[z_i \notin \mathcal{B}_\rho(\boldsymbol{s}_k) \text{ or } z_i \notin \mathcal{C}_0 \text{ or } f(\boldsymbol{z}_i) \ne 0\right] \tag{27}$$

$$\le \left(1 - \frac{\eta}{2N}\right)^{m_0} \tag{28}$$

$$\le \exp\left(-\frac{m_0 \eta}{2N}\right) \ge \frac{\delta}{2N}, \tag{29}$$

In the last line we used $1 + x \le e^x$, the fact that $m_0 = \frac{1}{4}\mu(C)m$, and the expression for $m$ from Theorem 2. Now we can use the union bound to prove the lemma:

$$\mathbb{P}\left[\text{ some } \mathcal{B}_\rho(\boldsymbol{s}_k) \text{ for } k \le K \text{ contains no } \boldsymbol{z}_i : i \le m_0, \ f(\boldsymbol{z}_i) \ne 0 \,\right] \le K \frac{\delta}{N} \le \delta. \tag{30}$$

∎

Finally, using both Lemma 8 and Lemma 10, we can finish:

$$\mathcal{R}_{\text{Adv},2\rho}(f, \mu) \ge \frac{1}{2}\mu(\mathcal{C})\, \mu_0 \left(\bigcup_{i \le m_0,\ f(\boldsymbol{z}_i) \ne 0} \mathcal{B}_{2\rho}(\boldsymbol{z_i})\right). \tag{31}$$

$$\ge \frac{1}{2}\mu(\mathcal{C})\, \mu_0 \left(\bigcup_{k=1}^{K} \mathcal{B}_\rho(\boldsymbol{s}_k)\right) \ge \frac{1}{4}\mu(\mathcal{C}). \tag{32}$$

□

## B  PROOF OF PROPOSITION 4

**Proposition 4.** *Let $\mu$ be the uniform distribution on $\mathbb{S}^{d-1} = \{x_1, \ldots, x_d \in \mathbb{R}^d : x_1^2 + \ldots + x_d^2 = 1\}$, and let the ground truth classifier $f^*$ be a threshold function on $x_1$: $f^*(x) = \mathbb{1}_{x_1 > \frac{1}{2}}$. Consider any adversarial radius $\rho < \frac{1}{4}$ in the Euclidean metric. Then, for any label noise $\eta < 1$: with high probability, there exists a classifier $f$ that interpolates $m = \lfloor 1.01^d \rfloor$ samples from the label noise distribution, such that $\mathcal{R}_{\text{Adv},\rho}(f, \mu) = o_d(1)$.*

*Proof.* Let the $m = 1.01^d \le \exp(d/80)$ samples be $\boldsymbol{z}_1, \ldots, \boldsymbol{z}_m$ with labels $y_1, \ldots, y_m \in \{0, 1\}$. Almost surely the $\boldsymbol{z}_i$ are distinct. Define the interpolating classifier $f : \mathbb{R}^d \to \{0, 1\}$ as

$$f(\boldsymbol{x}) = \begin{cases} y_i & \text{if } \boldsymbol{x} \in \{\boldsymbol{z}_1, \ldots, \boldsymbol{z}_m\}; \\ \mathbb{1}_{x_1 > \frac{1}{2}} & \text{otherwise.} \end{cases} \tag{33}$$

We want to show $f$ is robust. Draw $\boldsymbol{x} = (x_1, \ldots, x_d)$ uniformly on $\mathbb{S}^{d-1}$. There are only two ways $\boldsymbol{x}$ can contribute to the adversarial risk $\mathcal{R}_{\text{Adv},\rho}(f, \mu)$:

- $\boldsymbol{x}$ is close to a training sample $\boldsymbol{z_i}$ with label noise;

- $\boldsymbol{x}$ is close to the "decision boundary" $x_1 = \frac{1}{2}$ of $\mathbb{S}^{d-1}$.

Hence, remembering Equation (1),

$$\mathcal{R}_{\text{Adv},\rho}(f, \mu) \le \mathbb{P}\left[\boldsymbol{x} \text{ is in a } \rho\text{-ball around at least one of the } \boldsymbol{z}_i\right] + \mathbb{P}\left[\frac{1}{2} - \rho \le x_1 \le \frac{1}{2} + \rho\right]. \tag{34}$$

$$\le \mathbb{P}\left[\boldsymbol{x} \text{ is in a } \rho\text{-ball around at least one of the } \boldsymbol{z}_i\right] + \mathbb{P}\left[x_1 \ge \frac{1}{2} - \rho\right]. \tag{35}$$

By the union bound,

$$\mathbb{P}\left[\boldsymbol{x} \text{ is in a } \rho\text{-ball around at least one of the } \boldsymbol{z}_i\right] \tag{36}$$

$$\leq m\,\mathbb{P}\left[\|\boldsymbol{x} - \boldsymbol{z}_1\|_2 \leq \rho\right] \tag{37}$$

$$\leq m\,\mathbb{P}\left[\|\boldsymbol{x}\|^2 + \|\boldsymbol{z}_1\|^2 - 2\langle \boldsymbol{x}, \boldsymbol{z}_1 \rangle \leq \rho^2\right] \tag{38}$$

$$= m\,\mathbb{P}\left[\langle \boldsymbol{x}, \boldsymbol{z}_1 \rangle \geq 1 - \rho^2/2\right]. \tag{39}$$

As $\mu$ is rotationally invariant, $\langle \boldsymbol{x}, \boldsymbol{z}_1 \rangle$ is distributed the same as $x_1$. We have proved

$$\mathcal{R}_{\mathrm{Adv},\rho}(f, \mu) \leq m\,\mathbb{P}\left[x_1 \geq 1 - \frac{\rho^2}{2}\right] + \mathbb{P}\left[x_1 \geq \frac{1}{2} - \rho\right]. \tag{40}$$

We can bound $\mathbb{P}[x_1 \geq t]$ for $t > 0$ as follows: let $g_1, \ldots, g_d$ be i.i.d. standard $N(0,1)$ random variables.

$$\mathbb{P}\left[x_1 \geq t\right] = \mathbb{P}\left[\frac{g_1}{\sqrt{g_1^2 + \ldots + g_d^2}} \geq t\right] \tag{41a}$$

$$= \mathbb{P}\left[g_1^2 \geq t^2(g_1^2 + \ldots + g_d^2)\right] \tag{41b}$$

$$= \mathbb{P}\left[\frac{1 - t^2}{t^2} g_1^2 \geq g_2^2 + \ldots + g_d^2\right] \tag{41c}$$

$$\leq \mathbb{P}\left[\frac{1 - t^2}{t^2} g_1^2 \geq \frac{d-1}{2}\right] + \mathbb{P}\left[g_2^2 + \ldots + g_d^2 \leq \frac{d-1}{2}\right], \tag{41d}$$

where the last inequality is because $a \geq b$ implies $a \geq c$ or $b \leq c$.

As $0 < \rho < \frac{1}{4}$, we can take $t = \frac{1}{4}$ in both probabilities in Equation (40). We now use the often-cited chi-square bounds from Lemma 1 in Laurent and Massart (2000).

$$\mathbb{P}\left[g_2^2 + \ldots + g_d^2 \leq (d-1) - 2\sqrt{(d-1)s}\right] \leq \exp\left(-s\right) \tag{42a}$$

$$\mathbb{P}\left[g_1^2 \geq 1 + 2\sqrt{s} + 2s\right] \leq \exp\left(-s\right) \tag{42b}$$

Then for $s = \frac{d}{40}$, it's easy to see that both probabilities in Equation (41d) are less than the corresponding probabilities in Equation (42b) and Equation (42a).

Finally, as $d$ goes to infinity,

$$\mathcal{R}_{\mathrm{Adv},\rho}(f, \mu) \leq m\exp(-d/40) + \exp(-d/40) \tag{43}$$

$$\leq \exp(-d/80) + \exp(-d/40) \to 0. \tag{44}$$

$\square$

## C  POISONING THEOREM

Recall the definition of the adversarial risk $\mathcal{R}_{\mathrm{Adv},\rho}$ from Section 2:

$$\mathcal{R}_{\mathrm{Adv},\rho}(f, \mu) = \mathbb{P}_{\boldsymbol{x} \sim \mu}\left[\exists \boldsymbol{z} \in \mathcal{B}_\rho(\boldsymbol{x}),\ f^*(\boldsymbol{x}) \neq f(\boldsymbol{z}))\right]. \tag{45}$$

For interpolating classifiers with minimal test error, there are two sources of adversarial risk: decision boundaries and label noise. In this paper, we are specifically interested in the latter. The theorem is easiest to formalize in the case where the decision boundary contribution to the adversarial risk is negligible. For example, this is the case when the classes are separable with a large margin, or in real-world datasets when there is not much data which humans would label ambiguously.

Therefore, instead of working with the adversarial risk, we introduce the *separable proxy* for the adversarial risk. Let the label noised points be $\mathcal{S} = \{\boldsymbol{s}_1, \ldots, \boldsymbol{s}_N\}$, and let $f$ interpolate the training set, and otherwise minimize the test error.

$$\hat{\mathcal{R}}_{\mathrm{Adv},\rho}(\mathcal{S}, \mu) \stackrel{\mathrm{def}}{=} \mathbb{P}_{\boldsymbol{x} \sim \mu}\left[\exists 1 \leq k \leq N,\ \boldsymbol{s}_k \in \mathcal{B}_\rho(\boldsymbol{x})\right] \leq \mathcal{R}_{\mathrm{Adv},\rho}(f, \mu). \tag{46}$$

The proxy adversarial risk $\hat{\mathcal{R}}^{\text{Poison}}$ for the poisoned case is easy to define:

$$\hat{\mathcal{R}}^{\text{Poison}}_{\rho,N} = \sup_{\boldsymbol{s}_1,\ldots,\boldsymbol{s}_N \in \text{supp}(\mu)} \hat{\mathcal{R}}_{\text{Adv},\rho}(\{\boldsymbol{s}_1,\ldots,\boldsymbol{s}_N\}, \mu) \tag{47}$$

$$= \sup_{\boldsymbol{s}_1,\ldots,\boldsymbol{s}_N \in \text{supp}(\mu)} \mu\left(\bigcup_{k=1}^{N} \mathcal{B}_\rho(\boldsymbol{s}_k)\right). \tag{48}$$

In fact, by a compactness argument, the sup above can be replaced by $\max$, but this is not important for the theorem we want to prove.

The uniform version $\hat{\mathcal{R}}^{\text{Unif}}$ is a random variable. Its value depends on the random training set and which points get their labels flipped. We define it as follows:
Sample a training set of $T$ random points from $\mu$ independently, and let $\mathcal{S}$ be a random subset of the training set, with each point taken with probability $\eta$. Then

$$\hat{\mathcal{R}}^{\text{Unif}}_{\rho,T,\eta} := \hat{\mathcal{R}}_{\text{Adv},\rho}(\mathcal{S}, \mu). \tag{49}$$

We are now ready to state our formal theorem.

**Theorem 11.** *Let $\eta > 0$ be the label noise rate, and let $T, m$ be positive integers representing the training set sizes for Uniform and Poisoner. Fix $N = \lfloor \eta m \rfloor$ to be be the number of labels the poisoner can flip. For $\delta > 0$, with probability $1 - \delta$, for any $\rho > 0$,*

$$\hat{\mathcal{R}}^{\text{Unif}}_{2\rho,T,\eta} \geq \frac{1}{2} \hat{\mathcal{R}}^{\text{Poison}}_{\rho,N} \tag{50}$$

*for $T = \Omega\left(\frac{m(\log m + \log \frac{1}{\delta})}{\hat{\mathcal{R}}^{\text{Poison}}_{\rho,N}}\right)$.*

To be precise, the Uniform adversary takes at least the following number of samples:

$$T = \frac{2N}{\eta \hat{\mathcal{R}}^{\text{Poison}}_{\rho,N}} \left(\log N + \log \frac{1}{\delta}\right) = \frac{2m}{\hat{\mathcal{R}}^{\text{Poison}}_{\rho,N}} \left(\log m + \log \eta + \log \frac{1}{\delta}\right). \tag{51}$$

The proof is quite similar in spirit to the proof of Theorem 2.

*Proof.* Let the Poisoner pick points $\boldsymbol{s}_1, \ldots, \boldsymbol{s}_N \in \mathbb{R}^d$. We want to prove that with high probability,

$$\hat{\mathcal{R}}^{\text{Unif}}_{2\rho,T,\eta} \geq \frac{1}{2} \mu\left(\bigcup_{k=1}^{N} \mathcal{B}_\rho(\boldsymbol{s}_k)\right).$$

We show that, with high probability, a sample of $T$ points from $\mu$, with uniform label noise with probability $\eta$, will result in many of the balls $\mathcal{B}_\rho(\boldsymbol{s}_k)$ having a mislabeled point in them.

Order the points $\boldsymbol{s}_1, \ldots, \boldsymbol{s}_N$ such that

$$\mu\left(\mathcal{B}_\rho(\boldsymbol{s}_1)\right) \geq \mu\left(\mathcal{B}_\rho(\boldsymbol{s}_2)\right) \geq \cdots \geq \mu\left(\mathcal{B}_\rho(\boldsymbol{s}_N)\right). \tag{52}$$

By Lemma 8 with $\alpha = \frac{1}{2}$, we know that there exists $K$ such that

$$\mu\left(\bigcup_{k=1}^{K} \mathcal{B}_\rho(\boldsymbol{s}_k)\right) \geq \frac{1}{2} \mu\left(\bigcup_{k=1}^{N} \mathcal{B}_\rho(\boldsymbol{s}_k)\right) = \frac{1}{2} \hat{\mathcal{R}}^{\text{Poison}}_{\rho,N} \tag{53}$$

and $\mu\left(\mathcal{B}_\rho(\boldsymbol{s}_k)\right) \geq \frac{1}{2N} \hat{\mathcal{R}}^{\text{Poison}}_{\rho,N}$ for all $1 \leq k \leq K$.

As in Appendix A, we proceed to show that with high probability, a random sample of $T$ points from $\mu$ will hit each of the balls $\mathcal{B}(\boldsymbol{s}_k, \rho)$ for $k \leq K$, because the chosen balls are dense enough to get hit when $T \gtrsim m \log m$. This is enough to prove

$$\hat{\mathcal{R}}^{\text{Unif}}_{2\rho,T,\eta} \geq \mu\left(\bigcup_{k=1}^{K} \mathcal{B}_\rho(\boldsymbol{s}_k)\right), \tag{54}$$

since any two points in the same $\mathcal{B}_\rho(\boldsymbol{s}_k)$ are within distance $2\rho$ of each other.

Let $\boldsymbol{z}_1, \ldots, \boldsymbol{z}_T$ be a random dataset of $T$ points from $\mu$. We have a lemma similar to Lemma 10:

**Lemma 12.** *With probability $1 - \delta$, each $\mathcal{B}_\rho(s_k)$ for $k \leq K$ contains a mislabeled point $z_i$.*

*Proof.* We have

$$\mathbb{P}\left[z_i \in \mathcal{B}_\rho(s_k)\right] \geq \frac{1}{2N}\hat{\mathcal{R}}^{\mathrm{Poison}}_{\rho,N} \tag{55}$$

and because the label corruption is independent from everything, we also have

$$\mathbb{P}\left[z_i \in \mathcal{B}_\rho(s_k) \text{ and } z_i \text{ mislabeled}\right] \geq \frac{\eta}{2N}\hat{\mathcal{R}}^{\mathrm{Poison}}_{\rho,N}. \tag{56}$$

Therefore,

$$\mathbb{P}\left[\mathcal{B}_\rho(s_k) \cap \{\text{ mislabeled } z_i \} = \emptyset\right] \tag{57}$$

$$= \prod_{i=1}^{T} \mathbb{P}\left[z_i \notin \mathcal{B}_\rho(s_k) \text{ or } z_i \text{ mislabeled}\right] \tag{58}$$

$$\leq \left(1 - \frac{\eta}{2N}\hat{\mathcal{R}}^{\mathrm{Poison}}_{\rho,N}\right)^{T} \tag{59}$$

$$\leq \exp\left(-\frac{T\eta}{2N}\hat{\mathcal{R}}^{\mathrm{Poison}}_{\rho,N}\right) \leq \frac{\delta}{N}. \tag{60}$$

Here in the last line we used $1 + x \leq e^x$ and Equation (51).

Now we can use the union bound to prove the lemma:

$$\mathbb{P}\left[\text{ some } \mathcal{B}_\rho(s_k) \text{ for } k \leq K \text{ contains no mislabeled } z_i \right] \leq K\frac{\delta}{N} \leq \delta. \tag{61}$$

∎

Combining Equation (53) and Equation (54), we have the desired

$$\hat{\mathcal{R}}^{\mathrm{Unif}}_{2\rho,T,\eta} \geq \frac{1}{2}\hat{\mathcal{R}}^{\mathrm{Poison}}_{\rho,N}. \tag{62}$$

As this holds for any $s_1, \ldots, s_N$, and the left-hand-side does not depend on the chosen points $s_k$, we can take the supremum over all possible choices of $s_1, \ldots, s_N$ to prove the theorem. □

## D   PROOF OF PROPOSITION 6

**Proposition 6.** *Let $A < B$ be integers with $A$ much smaller than $B$. Let $\mu$ be a mixture model on $\mathbb{R}$ supported on a disjoint union of $A + B$ intervals, such that half of the mass is on the first $A$ intervals and half of the mass is on the last $B$ intervals:*

$$\mu = \frac{1}{2A}\sum_{i=1}^{A}\mathrm{Unif}\left(i, i + \frac{1}{2}\right) + \frac{1}{2B}\sum_{j=1}^{B}\mathrm{Unif}\left(A + j, A + j + \frac{1}{2}\right)$$

*Let the ground truth label be zero everywhere. Sample two datasets $\mathcal{D}_1, \mathcal{D}_2$ of size $m$ from $\mu$ using two different label noise distributions: For $\mathcal{D}_1$, flip the label of each sample $x \in [0, A + B]$ independently with probability $\eta$. For $\mathcal{D}_2$, flip the label of each sample $x \in [A, A + B]$ independently with probability $2\eta$, and leave the labels of the other samples unchanged. Then, for any $\rho, \delta \in \left(0, \frac{1}{2}\right)$, for the number of samples $m = \tilde{\Theta}_\rho(A)$ (ignoring log terms), we have that with probability $1 - \delta$:*

- *For any $f$ which interpolates $\mathcal{D}_1$, the adversarial risk is large: $\mathcal{R}_{\mathrm{Adv},\rho}(f, \mu) = \Omega_\rho(1)$.*

- *There exists $f$ which interpolates $\mathcal{D}_2$, such that $\mathcal{R}_{\mathrm{Adv},\rho}(f, \mu) = O_\rho\left(\frac{A}{B}\right)$.*

*Proof.* To prove the adversarial risk for $\mathcal{D}_1$, we simply invoke Theorem 2. Consider the compact $\mathcal{C} = \bigcup_{i=1}^{A} \left( i, i + \frac{1}{2} \right)$. The covering number for $\mathcal{C}$ is $N = \frac{A}{2\rho}$ and its probability mass $\mu\left(\mathcal{C}\right) = \frac{1}{2}$. By Theorem 2, for $m \geq \frac{16A}{\rho\eta} \log \left( \frac{A}{\rho\delta} \right)$, with probability greater than $1 - \delta$ we have that $\mathcal{R}_{\mathrm{Adv},\rho}(f, \mu) \geq \frac{1}{8}$. This proves the first part.

For the second part, using Hoeffding's inequality, as long as $m \geq 16 \log \left( \frac{2}{\delta} \right)$, we have that with probability at least $1-\delta$, the number of samples in $[A, A + B]$ is less than $\frac{3m}{4}$. Therefore, the number of mislabeled samples in that interval is also less than $\frac{3m}{4}$ with the same probability. Now consider the interpolator that is zero everywhere except at the mislabeled points. The maximum adversarial risk of this interpolator is the probability mass of the union of the intervals $[A + j, A + j + \frac{1}{2}]$ each of the mislabeled points lie in. This probability mass is at most $\frac{3m\rho}{8B}$. Setting, $m = \frac{16A}{\rho\eta} \log \left( \frac{A}{\rho\delta} \right)$, we obtain $\mathcal{R}_{\mathrm{Adv},\rho}(f, \mu) = \widetilde{O}_\rho \left( \frac{A}{B} \right)$. $\qquad\square$

## E    PROOF OF INDUCTIVE BIAS

**Theorem 7.** *For any $\rho > 0$, there exists a distribution $\mu$ on $\mathbb{R}^2$ and two hypothesis classes $\mathcal{H}$ and $\mathcal{F}$, such that for any label noise rate $\eta \in (0, 1/2)$ and dataset size $m = \Theta\left(\frac{1}{\eta}\right)$, in expectation we have that: for all $h \in \mathcal{H}$ that interpolate $S_{m,\eta}$,*

$$\mathcal{R}_{\mathrm{Adv},\rho}\left(h, \mu\right) \geq \Omega\left(1\right); \tag{10}$$

*whereas there exists an $f \in \mathcal{F}$ that interpolates $S_{m,\eta}$ and $\mathcal{R}_{\mathrm{Adv},\rho}\left(f; \mu\right) = \mathcal{O}\left(\rho\right)$.*

*Proof.* For any $\rho \geq 0$, $W \gg \rho$, construct a distribution $\mu$ on $[0, W] \times \{0\}$ as follows. Distribute the covariates uniformly randomly in $[0, \frac{W}{2} - 2\rho] \bigcup [\frac{W}{2} + 2\rho, W]$ and then label then with the ground truth labelling function $f^*\left(x\right) = \mathbb{1}\{x_1 \geq \frac{W}{2}\}$ where $x = [x_1, x_2]$ is the two-dimensional covariate. Next, we construct an $m$ dimensional dataset and flip each label independently with probability $1 - \eta$. We denote this set with $S_{m,\eta}$.

The hypothesis class $\mathcal{F}$ is the class of one-dimensional thresholds on the first coordinate of the input space (ignores the second coordinate entirely). Define the following interpolating classifier $f \in \mathcal{F} : \mathbb{R}^2 \to \{0, 1\}$ as follows

$$f(x) = \begin{cases} y_1 & \text{if } x \text{ is in } S_{m,\eta} \\ \mathbb{1}\{x_1 \geq \frac{W}{2}\} & \text{otherwise} \end{cases}.$$

As the sampling of the covariates and the label noise are independent events,

$$\mathbb{E}_{S_{m,\eta}} [\# \text{ of mislabeled points in } S_{m,\eta}] = m\eta.$$

Then the expected measure of the set of points adversarially vulnerable by an adversary of perturbation magnitude $\rho$ on the classifier $h$, as defined above, is upper bounded by $2\rho m\eta$. Using the fact that the total measure of the domain is $W$ and that $m = \Theta\left(\frac{1}{\eta}\right)$, we get that

$$\mathbb{E}_{S_{m,\eta}} [\mathcal{R}_{\mathrm{Adv},\rho}\left(f; \mu\right)] \leq \frac{2\rho m\eta}{W} = \mathcal{O}(\rho).$$

Next, consider the hypothesis class $\mathcal{H}$ defined as follows. Given a set of points $\mathcal{Z} = \{z_1, \ldots, z_k\} \in [0, W]^k$ and $\gamma > \rho$, define the hypothesis

$$h_{\mathcal{Z},\gamma}\left(x\right) = \begin{cases} 1 & \exists z \in \mathcal{Z} \mid \mathbb{1}\{x_2 < \rho\} \wedge \mathbb{1}\{x_1 = z\} \\ 1 & \exists z \in \mathcal{Z} \mid \mathbb{1}\{x_2 < \rho\} \wedge \mathbb{1}\{|x_1 - z| \leq \gamma\} \\ 0 & \text{otherwise.} \end{cases}$$

If $\widetilde{S}$ is the set of mislabeled 1s in $S_{m,\eta}$, then for any interpolating classifier $h_{\mathcal{Z},\gamma}$, it holds that $\widetilde{S} \subseteq \mathcal{Z}$. Next, by construction, for every point $z \in \mathcal{Z}$, it holds that all points $x \in [z - \gamma, z + \gamma]$ can adversarially perturbed in the $x_2$ component to obtain the label 1. Thus the total measure of the

adversarially vulnerable set of points is greater than the number of mislabeled points, whose original label is zero, multiplied with $2\gamma$, which is $2m\eta\gamma$.

Thus, we have that for any $h \in \mathcal{H}$ that interpolates $S_{m,\eta}$,

$$\mathbb{E}_{S_{m,\eta}} \left[ \mathcal{R}_{\mathrm{Adv},\rho}\left(f;\mu\right) \right] \geq \min \left( \frac{2\gamma m\eta}{W}, \frac{1}{2} \right) = \Omega(\gamma).$$

$\square$

We proved the adversarial risk bounds only in expectation over the training set; but note that both of the bounds in Theorem 7 can be transformed into high probability bounds using concentration inequalities.

For simplicity, we did not treat the second bound in the theorem as a learning problem. However, it is possible to show that there exists a learning algorithm that uses a similar number of samples to output $f \in \mathcal{F}$ such that the adversarial risk is $\mathcal{O}\left(\rho\right)$.

## F  SEPARATION AND AVERAGE DISTANCES OF DIFFERENT CLASSES

For a classification dataset, it is not easy to check if a given class can be covered by balls of some radius in a given metric, such that each ball contains only points from a single class.

However, we can estimate the distance distribution between points inside a class and between points of different classes, If those two distributions are similar, and especially if the minimum distances are roughly the same, then it seems likely that it's not possible for the balls to contain only points from a single class.

The plots in Figure 6 strongly suggest that the points of different classes are not extremely "far off" in the $\|\cdot\|_2$ or $\|\cdot\|_\infty$ metrics, compared to the distances between points inside a class.

## G  EXPERIMENTS REGARDING INDUCTIVE BIAS

We show that the the structure of the T-shaped classifier used in Theorem 7 is visible when fitting neural networks to label noise on a tiny dataset. We sample a three dimensional dataset of points $(X, Y, Z)$ with labels in $\{0, 1\}$ as follows:

- $X$ is sampled uniformly from the segment $[0, 1]$;
- $Y$ is sampled from a normal distribution $\mathcal{N}(0, 0.1)$;
- $Z$ is sampled from a normal distribution $\mathcal{N}(0, 0.001)$;
- The label is 1 if $X > 0.5$, and 0 otherwise.

We sample 50 points from this distribution to create the clean dataset. To create the noisy dataset, we randomly flip $10\%$ of the labels to generate the noisy dataset.

Then we train a one-hidden layer MLP with 1000 hidden units using the ADAM optimizer with a learning rate of 0.01. The decision boundary after running this for 350 epochs with a batch size of 20 is plotted in Appendix G. All our models interpolate the dataset (both clean and noisy).

We plot the decision region in the $XY$ plane for three different values of the Z dimension in Appendix G for models trained on the noisy dataset as well as the clean dataset. The first row in a box corresponds to the model trained on the noisy dataset and the second row corresponds to the model trained on the clean dataset. The maroon circles inside the plots are balls of radius 0.04, drawn around the points with label noise, indicating the region of adversarial vulnerability induced by the points in that plane.

As visualizations like these are often susceptible to variance due to random seeds, we report for three different seeds, denoted as Run 1, Run 2, and Run 3.

To interpret the T-like structure from Figure 7 in these experiments, note that $\rho = 0.04$, so the "head" of the 'T' is in the $XY$ plane for $Z = 0.04$. Further, the $Z = -0.04$ is essentially the head

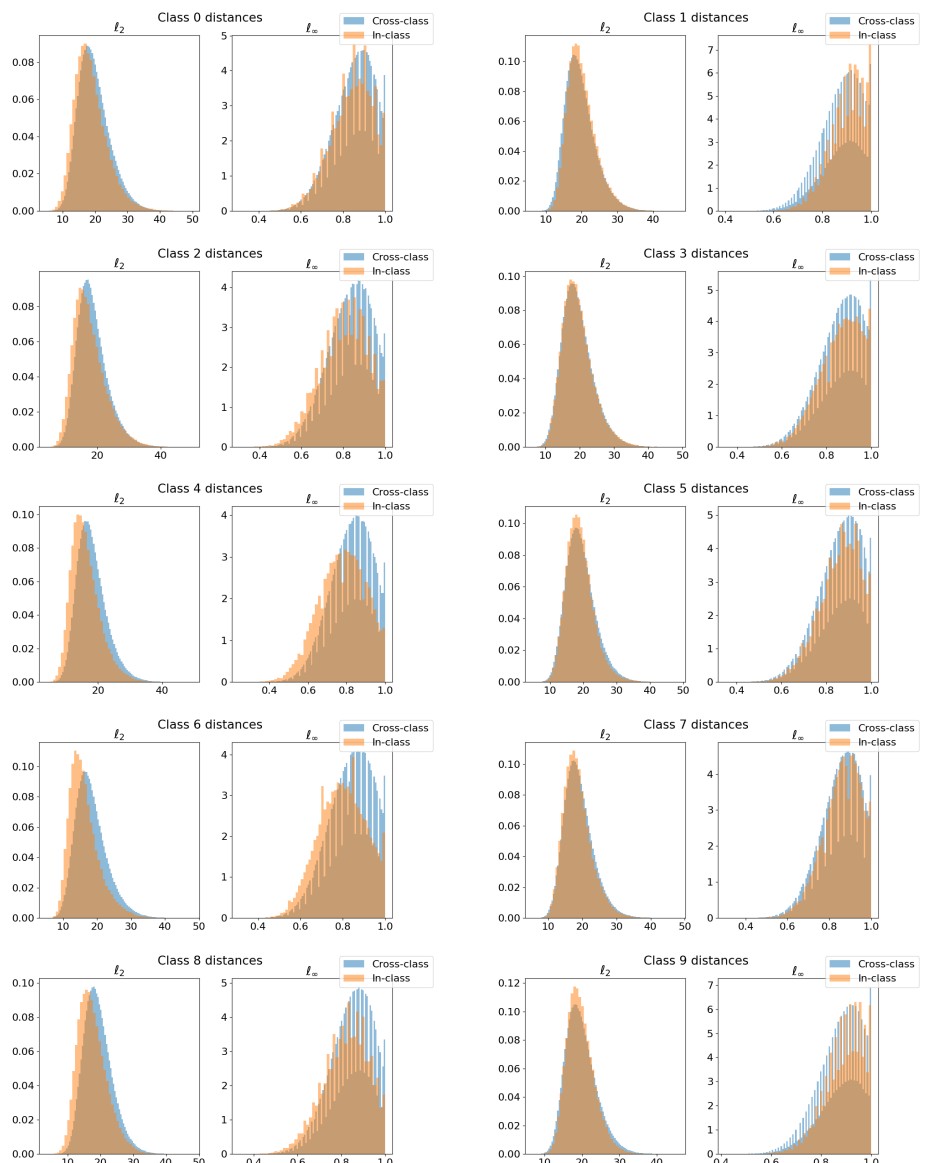

Figure 6: Separation and average distances of different classes in CIFAR-10.

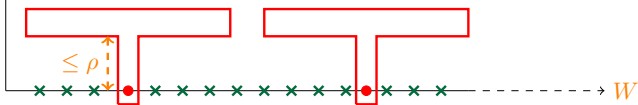

Figure 7: Visualization of a portion of the distribution $\mu$ and the hypothesis class $\mathcal{H}$ used in Theorem 7. The crosses are the mislabeled examples and the circles are correctly labelled examples. All the circles are adversarially vulnerable to upwards perturbations of magnitude less than $\rho$.

of the 'T' for the other class. In the first rows in all of the boxes (i.e. the noisy dataset), note that "heads of the Ts" are almost entirely within the decision region of one of the classes. This indicates that all points on the $XY$ plane at $Z = 0$ can be perturbed along the $Z$-axis with a perturbation less than $0.04$ to change its predicted label, yielding an adversarial risk of 100%. However, in the $Z = 0$ plane, the region of vulnerability is the union of the maroon balls, which is significantly smaller than what is what induced by perturbation along the Z-dimension. This suggests that our intuition with the T-shaped classifiers may be relevant in practice for neural networks.

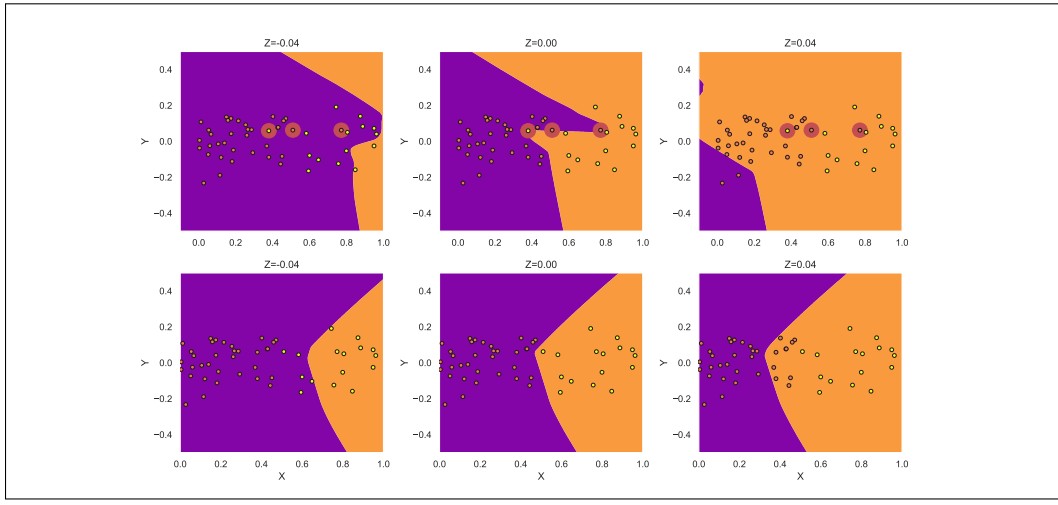

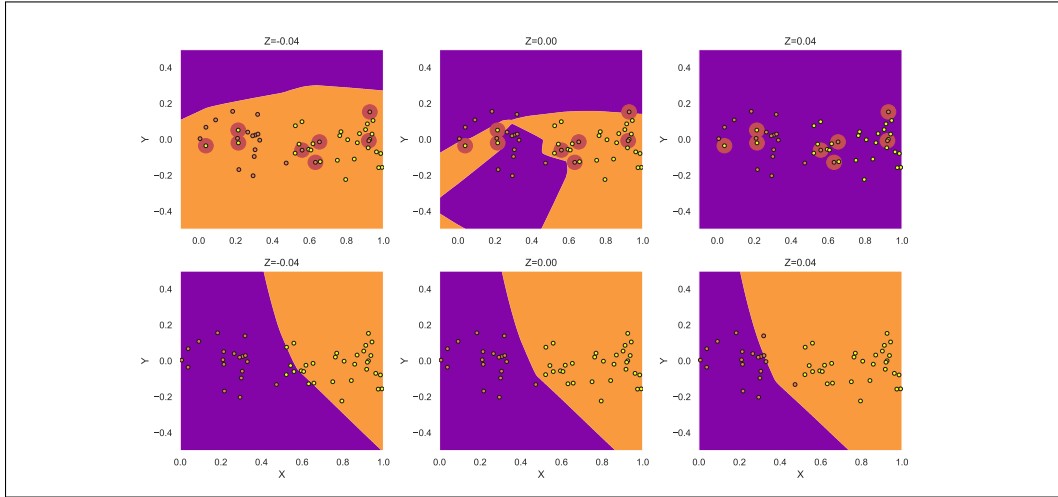

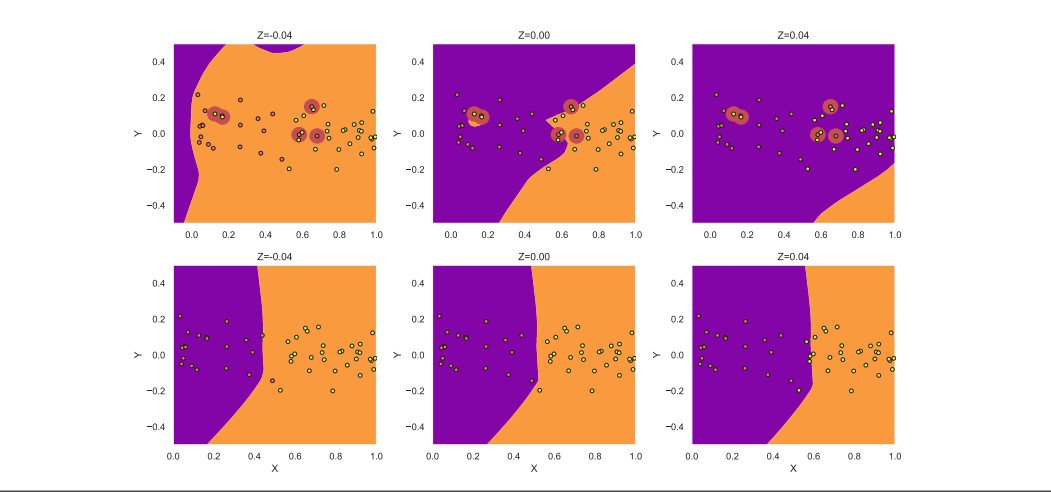

Figure 8: Each box is a different independent sample of the dataset. The first row in each box is with label noise, and the second row is without label noise. The three plots in each row ($Z \in \{-0.04, 0.00, 0.04\}$) show the decision boundary of the interpolating model on the $XY$ plane for different values of $Z$. The $Z = 0.04$ can be interpreted as the head of the 'T' shaped decision region and $Z = -0.04$ is similarly an inverted 'T' for the other class. The plots clearly show that when the model interpolates label noise, the width of the head of the 'T's are significantly more responsible for adversarial vulnerability than the decision region in the $Z = 0$ plane.

