# OpenReview forum: "A law of adversarial risk, interpolation, and label noise"
_ICLR.cc/2023/Conference — ICLR 2023 poster_

### Official Review · Reviewer_UsFF · 2022-10-16

**Confidence:** 3
**Correctness:** 3
**Technical Novelty And Significance:** 4
**Empirical Novelty And Significance:** 4
**Recommendation:** 8

**Clarity, Quality, Novelty And Reproducibility:**

In fact, I do not fully check the proofs. However, the theoretical results from this submission are novel. This submission should be polished, especially the contributions and conclusion.

The authors do not provide code. Thus the reproducibility cannot be verified. Especially the reproduction of Figure 3-related experiments.

**Strength And Weaknesses:**

Strength:

1. This paper is very interesting. The authors provide a new adversarial risk when data contains label noise. Theorem 2 is more involved than the results from Sanyal et al. (2021) when the assumption of 'Each ball only contains points from a single class' does not hold.

2. I like Figure 3, it is clear to show that the uniform noise is worst than human-manipulated noise.

3. The motivation of this paper is clear, which is to improve and enhance the theoretical results from Sanyal et al. (2021).

Weaknesses:

1. I expect the authors can provide a section to discuss related works. For example, the work [Zhu, Jianing, et al. (2021)] should be discussed in this submission because it is related to label noise and adversarial training.

Zhu, Jianing, et al. "Understanding the interaction of adversarial training with noisy labels." arXiv preprint arXiv:2102.03482 (2021).

2.  I also expect the authors can provide a clear claim about contributions at the end of Section 1, which can make this submission clear. Furthermore, a conclusion section should be added.

**Summary Of The Paper:**

This submission provides a theoretical result of a lower bound of adversarial risk when data contains noise. Experiments on MNIST and CIFAR10 also demonstrate the correctness of the fact that uniform label noise is more harmful than typical real-world label noise. The authors also show the connection between inductive biases and label noise.

**Summary Of The Review:**

In general, I am not familiar with the theoretical parts of this topic. However, I suggest accepting this paper. The results of this paper can provide more insights for researchers working on the area that from label noise and adversarial training.

---

> ### Author Response · Authors · 2022-11-13
> **Reply to reviewer UsFF**
>
> We thank the reviewer for their detailed reading, thoughtful comments, references, and overall positive assessment of our work.
>
> **Regarding discussion on contribution, conclusion, and related work**
>
> We thank the reviewer for pointing out this, indeed relevant, work that we had missed. We have now added the discussion of Zhu et. al. 2021 in our Introduction in page 1 (in blue). Due to space shortage, we have not included individual sections on contribution, conclusion, and related work but we have tried to merge them into the existing sections. We can perhaps change the overview section to reflect the contributions more clearly and if the reviewer suggests, we can change the last section on the conjecture to include a conclusion section.
>
> We follow standard experimental settings with pre-existing datasets and architectures. We were a bit overwhelmed with receiving so many reviews and doing new experiments. So, we plan to release the code soon, possibly after the review period is over.

---

### Official Review · Reviewer_BqSH · 2022-10-25

**Confidence:** 4
**Correctness:** 4
**Technical Novelty And Significance:** 4
**Empirical Novelty And Significance:** Not applicable
**Recommendation:** 8

**Clarity, Quality, Novelty And Reproducibility:**

Quality: their work consider an interesting problem -- understanding the role that fully interpolating data with label noise has on adversarial robustness. Their work is theoretically sound and even has practical significance considering the well known phenomenon of overfitting using deep neural networks.

Novelty: to my knowledge, these results are novel and include a significant improvement over previous work in this area. They include a comparison to older work, and I believe that their results are much more compelling and elegant.

Clarity: their work is well written -- their theorem statements and proof ideas are all easy and natural to follow. My only negative comment is addressed above where I claim that this paper could improve from a better explanation of "inductive bias."

**Strength And Weaknesses:**

This paper provides a simple, elegant result that cleanly describes the relationship between interpolating label noise and the resulting robustness. I particularly liked the proof intuition they included in their main body for most of their theoretical results.

I have few comments about this paper. First, I think it would be interesting if they connected this result to some sort of baseline result about the effect that fully interpolating noise has on standard classification. While such a result would probably require some sort of assumption about the hypothesis class being used (my understanding for their result is that by virtue of the nature of robustness, no assumptions are necessary on the classifier), I do think it would still serve for a meaningful comparison.

Secondly, I found their section regarding inductive bias a bit difficult to read. I think including a formal definition of ``inductive bias", or alternatively defining some sort of related entity would greatly strengthen this section. From my current understanding, it seems as though the reader is expected to connect Theorem 7 to the concept of inductive bias implicitly, and I think making this connection explicit would improve the presentation of this paper.

Finally, I wonder if anything can be said about classifiers that don't fully interpolate the entire dataset, but rather get as close to doing so as possible. In particular, this would describe situations in which the hypothesis class isn't broad enough to achieve 0 loss, but is nevertheless able to significantly overfit to noise.

**Summary Of The Paper:**

Let $S$ be a training set from data distribution $\mu$ that is labeled by baseline classifier $f^*$ with label noise $\eta$. This paper investigates the effect that fully interpolating over $S$ has on the adversarial robustness of the resulting classifier (with respect to $(\mu, f^*)$). They give a highly general result that lower bounds the adversarial loss based on the measure of a compact set, $\mathcal{C}$, along with the number of balls of radius $\frac{\rho}{2}$ needed to cover it. Their result consequently formalizes a simple intuition -- interpolating noise results in classifiers that change their values a lot, and this leads to poor robustness. To complement this, they show that in some cases, their bound on the amount of data needed for this phenomenon to occur is relatively tight.

They then discuss several implications of their results, including a comparison of the effect of noise with the effect of data poisoning as well a discussion of inductive bias. In all cases, they include a theoretical result (with proof) that gives some sort of lower bound on the robust loss of a classifier that interpolates a noisy training set. They then conclude by conjecturing that a similar phenomenon holds for neural networks, with the necessary sample size being related to the dimension as well as the number of neurons.

**Summary Of The Review:**

This is a good paper and should be accepted. It studies a simple but interesting problem thoroughly and elegantly. I believe this work is easily understandable and is nevertheless quite non-trivial.

---

> ### Author Response · Authors · 2022-11-13
> **Reply to Reviewer BqSH**
>
> We thank the reviewer for their detailed reading, thoughtful suggestions and comments, and overall positive assessment of our work. We would also like to highlight that the summary of the paper provided by the reviewer aligns with the message we wished to convey with the paper; we take this as a positive sign regarding the presentation of the paper. The reviewer has made some interesting comments, which we hope we can address in this rebuttal.
>
> **Connected this result to some sort of baseline result about the effect that fully interpolating noise has on standard classification**
> This question is precisely the topic of investigation in the subfield of _''benign overfitting’’_ and/or _"`harmless interpolation’’_. As pointed out by the reviewer, one of the assumptions required for this is precisely some assumption on the function class. For example, this is covered  in the case of linear regression in [1,2], for binary max margin classification in [3,4] and more recently in settings with stronger structure (like sparsity) [5,6]. We have already referred to some of them in our introduction, we will include some more discussion and references to these works  in our paper.
>
>  **Connect Theorem 7 to the concept of inductive bias implicitly**
>
> We have now included an appendix (Appendix G) where we provide experimental results to connect theorem 7 and its illustration in Figure 4 to the actual decision region of realistic experiments using neural networks. We hope this helps clarify the reference to inductive bias. However, we would also like to note that characterising inductive bias is a longstanding problem without a precise definition. Our theorem 7 provides an example and Appendix G shows that the example in theorem 7 is not very far from what is observed in neural networks. We included a slightly more detailed discussion in the general comments.
>
> **Anything can be said about classifiers that don't fully interpolate the entire dataset, but rather get as close to doing so as possible?**
>
> This is also another interesting question raised by the reviewer, which we think this can be an interesting future work. However, for full disclosure, we think the proof technique used in this manuscript will not yield any result for the setting where the dataset is not interpolated. In particular, we believe no lower bound can be provided on the adversarial error of such a classifier without further assumptions. To see why, note that the perfect classifier (i.e. the ground truth) cannot be distinguished from such a classifier (that satisfies the mentioned setting) by just considering the predicted label.  So, to get results for that setting, it perhaps needs some new insights and assumptions
>
>
>  [1] Muthukumar, Vidya, et al. "Harmless interpolation of noisy data in regression." IEEE Journal on Selected Areas in Information Theory 1.1 (2020): 67-83.
> [2] Bartlett, Peter L., et al. "Benign overfitting in linear regression." Proceedings of the National Academy of Sciences 117.48 (2020): 30063-30070.
> [3] Wang, Ke, and Christos Thrampoulidis. "Binary classification of Gaussian mixtures: abundance of support vectors, benign overfitting and regularization." arXiv preprint arXiv:2011.09148 (2020).
> [4] Chatterji, Niladri S., and Philip M. Long. "Finite-sample Analysis of Interpolating Linear Classifiers in the Overparameterized Regime." J. Mach. Learn. Res. 22 (2021): 129-1.
> [5] Donhauser, Konstantin, et al. "Fast rates for noisy interpolation require rethinking the effects of inductive bias." arXiv preprint arXiv:2203.03597 (2022).
> [6] Wang, G., et al.  Tight bounds for minimum $\ell_1$-norm interpolation of noisy data. Proceedings of The 25th International Conference on Artificial Intelligence and Statistics, 2022.

---

### Official Review · Reviewer_AQu9 · 2022-10-27

**Confidence:** 4
**Correctness:** 3
**Technical Novelty And Significance:** 2
**Empirical Novelty And Significance:** 3
**Recommendation:** 6

**Clarity, Quality, Novelty And Reproducibility:**

The paper is generally well-written, and seems reproducible from the proofs. There are novel ideas in the paper, but some seem to be incremental. Here are some points on improving the exposition:
- In the theorem statements, it would be helpful to remind what $f$ is.
- In Fig 2, ensure that the labels are not covering the plots
- A conclusion section would greatly help the paper to condense the key takeaways.
- The bar chart is a little confusing, would be good to plot the two with their averages and standard deviations (box and whisker plot).
- Define $\mathcal{H}$ & $\mathcal{F}$ in Theorem 7

**Strength And Weaknesses:**

**Strengths**:
- The paper is generally well-written and presents several angles on the underlying problem
- The discussion and experiments on long-tail label noise were pretty interesting to me, including the use of the memorization score.
- The authors are honest about the limitations of their theorems and explore potential ways to improve them to bring them closer to practice.

**Weaknesses**:
- I might be missing something, but I am not completely convinced that Theorem 2 is significantly better than Theorem 1 which is already known. Intuitively, covering number is essentially quantifying the finite set of balls needed to cover the space. In fact, in order to prove Theorem 2, the authors do prove the condition assumed in Theorem 1 with different constants. A discussion on the differences is very important. Furthermore, in the light of Section 3, it is unclear to me as to what the value of Theorem 2 is.
- The technical contribution is not super clear, and in the attempt to address different aspects of the problem, no particular one is explored deeply.

**Summary Of The Paper:**

The paper studies the role of label noise in the data in increasing adversarial risk of the trained classifier. The main theorem gives a constant lower bound on the adversarial risk of an arbitrary  interpolating classifier (one that achieves zero train error) trained on a noisy dataset of sufficiently large size. Subsequently, the authors argue how this result that allows arbitrary interpolating classifiers is weak to explain this phenomenon in practice since it requires very large samples. They give a constructive argument saying that allowing arbitrary interpolators necessarily leads to an exponentially large sample complexity requirement, showing that their theorem is tight. They also show how inductive biases can reduce this sample complexity requirement. Furthermore, the paper explores other noise models other than uniform noise and show that (1) uniform noise is as challenging as data poisoning attacks, (2) long-tail noise is more benign.

**Summary Of The Review:**

The paper has interesting ideas about the role of label noise such as the impact of the distribution of noise on the adversarial risk. However, the overall technical significance of the results is unclear to me. The main message of the paper gets lost in the various directions considered here. Therefore, I’m leaning slightly towards rejecting the paper. I would be happy to raise my score if the authors can (1) convince me on the technical novelty of the paper, or (2) explore practical implications of their results.

---

> ### Author Response · Authors · 2022-11-13
> **Reply to Reviewer AQu9**
>
> We thank the reviewer for the thoughtful suggestions and comments, the highlighted strengths, and especially on the actionable advice. We implemented some of the exposition improvements and will implement the others in the final version.
>
> Below, we provide comments that should help answer the specific questions:
>
> **Regarding Theorem 2 and Section 3**
> * As we have discussed in greater detail in our general comments, the point of Proposition 4 is to prove that this line of work (proving lower bound on adversarial risk for **any** interpolant) cannot hope to achieve better results than ours. Thus Sections 2 and 3, together, provide an upper bound and a matching lower bound. Our message here is:
> _“What we proved in this paper are the best possible results in this theoretical paradigm. However, the guarantees do not seem strong enough to explain the label noise <-> adversarial risk dependence seen in practice, unless we can make theorem 2 tighter by solving the hard optimization problem in Eq (8). Hence future research on this topic must adopt inductive bias assumptions.”_
>
> * We also argue below with **new experiments in Appendix F**, why we consider the assumptions in Theorem 1, which we get rid of in Theorem 2, to be overly constraining:
>     * The two assumptions in Theorem 1 imply that, in $\ell_{\infty}$ or $\ell_2$ distance there are regions in the input space, which contain data points of the same label (assumption 1) and these high density regions contain no points of the opposite label.
>     * A way to validate this would be to see whether there is a significant difference in the distribution of distances between points of the same label and those of the opposing label. We plot the density functions of those distributions in Figure 6 in the appendix, and show that the difference is small.
>
> * Note also that Theorem 2 is a bit stronger than Theorem 1 even when the assumptions of Theorem 1 hold. Imagine a distribution containing $N$ disjoint balls, and draw $m = \Omega(N / \eta c)$ samples, for some constant $c$. Then Theorem 1 requires each of the balls to have probability mass on the order of $c$, while Theorem 2 only requires the average ball density to be on the order of $c$.
>
> **Contribution of Theorem 5**
>
> In addition, we believe that Theorem 5 is also one of the key contributions in the paper.
> The uniformity of the label noise is a priori a very strong assumption and well known to be not reflected in real world data, which is nonetheless often the default assumption in the literature.
> We think showing that the “uniform” and “worst-case” label noise cases are conceptually similar is novel and technically significant.
>
> **Practical implications of inductive bias in Theorem 7**
>
> In Appendix G, we have included **new experiments  with neural networks to show that this behavior, with T-shaped decision boundary,  is a toy example similar to what is observed by neural networks**. The reason why the T-shaped decision regions in Theorem 7 amplify the effect of label noise to adversarial risk is that the decision boundaries induced by label noise are not orthogonal to the data manifold. **There is empirical evidence [1,2,3] to believe that adversarial vulnerability for neural networks are at least partly due to the same underlying reason as Theorem 7**.  While these papers do not consider label noise, they do support the “inductive bias” that leads to large adversarial error. We show this phenomenon more explicitly  by plotting the decision boundaries for neural networks that interpolate label noise in Appendix G.
>
>
> To conclude, we would like to stress that we consider our main theoretical results to be captured in Theorem 2, Corollary 3, Proposition 4, and Theorem 5. The later results Proposition 6 and Theorem 7 are an addition, where we show how certain assumptions can be used to improve the results, and we provide experimental validations of those assumptions in Figure 3 and Appendix G respectively. Please see the general comments for a complete picture of our view on the contributions.
>
> **We hope the above and the points mentioned in our general comments have helped answer your questions regarding our work, and hope you will consider increasing your score. If you have any further questions or suggestions we would be more than happy to engage in answering them.**
>
>
>
> [1] Stutz, David, Matthias Hein, and Bernt Schiele. "Disentangling adversarial robustness and generalization." Proceedings of the IEEE/CVF Conference on Computer Vision and Pattern Recognition. 2019.
> [2] Shamir, Adi, Odelia Melamed, and Oriel BenShmuel. "The dimpled manifold model of adversarial examples in machine learning." 2021.
> [3] Ma, Xingjun, et al. "Characterizing adversarial subspaces using local intrinsic dimensionality."  International Conference on Learning Representations. 2018.

---

> > ### Comment · Reviewer_AQu9 · 2022-11-16
> > **Response to the Rebuttal**
> >
> > Thank you for addressing my questions and concerns. Here are some comments:
> > - I understand the improvement that your theorem has, thanks for the clarification. I still do not think this improvement is technically significant compared to the prior result. As for the experiment, measuring the distribution of inter-class and intra-class distance is not sufficient cause the geometry matters. It might be better to estimate the probability of a point of the same class lying in a ball around each point.
> > - I agree that Theorem 5 is novel and interesting with a theoretical implication that uniform noise is not a good model if we hope to get any robustness. However, it is not immediately clear to me what the practical implication is.
> > - The new experiments in Appendix G are nice, however, they are still pretty limited since the data generation process is still in a very toy setting.
> >
> > Since the authors have partially addressed my 2 concerns, I'm willing to increase the score by 1. The authors need to put in more thought on the significance/implications of these results, and discuss in the paper.

---

### Official Review · Reviewer_59Qr · 2022-10-30

**Confidence:** 3
**Clarity, Quality, Novelty And Reproducibility:** The paper is clearly written and the …
**Correctness:** 4
**Technical Novelty And Significance:** 3
**Empirical Novelty And Significance:** 3
**Recommendation:** 6

**Strength And Weaknesses:**

Strengths:

- The first contribution of this work is Theorem 2, which improves the results of Sanyal et al. (2021). Specifically, the authors avoid the unwieldy assumptions from Sanyal et al. (2021) by using a more elegant concept called the covering number.  This not only characterizes data distribution properly but also gives a slightly stronger guarantee.
- The second contribution of this work is a critical discussion of the required sample size for Theorem 2. In particular, the authors show that the sample size for Theorem 2 is tight in the worst case, while the sample size of standard datasets such as MNIST is much smaller than that required by Theorem 2. Thus, it is not possible to obtain tighter bounds without further assumptions on the data or the model. The authors further demonstrate this point with specific examples.
- The third contribution of this work is that the authors prove that uniform label noise is on the same order of harmfmul as worst case data poisoning, given a slight increase in dataset size and adversarial radius. This result is pleasant on its own, while the connection between this one and the former two is obscure.

Weaknesses:

- Probably it is not surprising to conclude that label noise increases adversarial error. It is well known that the test error increases with increasing label noise (also shown in Figure 2). Since the adversarial error is an upper bound of the test error, it is natural to conclude that the adversarial error will increase accordingly.
- Theorem 2 only characterizes the amount of adversarial error, but fails to show why the adversarial error grows faster than the test error. I thought the latter is the most interesting phenomenon shown in Figure 2.
- Adversarial training is a popular method to minimize adversarial risk. It would be better if the theory can show the relationship between label noise and adversarial risk for adversarially trained classifiers (i.e., classifiers that correctly and robustly interpolate the training set).

**Summary Of The Paper:**

The work studies the relationship between label noise and adversarial risk for interpolating classiﬁers (i.e., classifiers with zero training error). The authors prove that interpolating label noise induces high adversarial risk for any data distribution when the sample size is very large. To better align the undesirable gap between the large sample size required by the theorem and the relatively small sample size of standard datasets, the authors argue that this requires further restrictions/assumptions on the data distribution or the inductive bias of the function class. Additionally, the authors prove that uniform label noise induces nearly as large an adversarial risk as the worst poisoning with the same noise rate, which is of interest on its own.

**Summary Of The Review:**

This paper provides a solid contribution to the theoretical understanding of the relationship between adversarial risk, interpolation, and label noise. It would be more impactful if the weaknesses listed above could be addressed well.

---

> ### Author Response · Authors · 2022-11-13
> **Reply to reviewer 59QR**
>
> We thank the reviewer for their careful reading of the paper, helpful comments, clarificatory questions, and overall positive assessments of the paper. Below, we provide answers to the specific questions of the reviewer and hope it helps alleviate the doubts.
>
> **Adversarial error is a upper bound on test error and test error increases with label noise**
>
> We would like to highlight two points in regards to this
> * **Test error may be small for interpolators:** As reviewer BqSH has pointed out, label noise affects the test error very differently to how it affects adversarial error in this paper. In fact, several recent papers have shown that it is possible to interpolate label noise without increasing test error significantly [1,2,3,4,5,6]. This depends on the function class, optimisation procedure etc and is widely studied in the subfield of _benign overfitting_ and _harmless interpolation_.
> * **Our results are model agnostic and apply even when test error is small:** Our results hold for all function classes and hence it applies even when test error is small, possibly arbitrarily close to zero. In short, even when the test error is very small for interpolators (which is possible), the adversarial error is necessarily large (as shown by our results).
> Thus, the intuition that test error increases with label noise and thus adversarial also increases does not apply to our setting. Here, adversarial error increases for all interpolators irrespective of whether and how fast the test error increases.
>
> **How fast the adversarial error increases compared with the test error**
>
> As pointed out in the previous question, answering this question requires making assumptions on the function class. For example, consider the setting of Corollary 3 with distribution $\mu$ and let $f$ be a classifier that obtains zero classification error on $\mu$.
>
>  Let $S_m$ be a dataset with $m$ training points where $m$ satisfies the condition of Corollary 3. Let $g$ be a classifier that classifiers a point $x$ as follows. If $(x,y)\in S_m$ then $g(x)=y$ else $g(x)=f(x)$. Note that $g$ also obtains zero test error as the set of noisy points in $S_m$ ($(x,y)\in S_m$ where $y\neq f(x)$) has measure zero. However, by theorem 2, the adversarial error is at least $0.25$.
>
> Thus, answering this question requires making further assumptions on $g$, which we think could be an interesting future work and would require  combining works like [1,2,3,4,5,6] with our work.
>
> **Showing bounds for AT trained classifier**
>
> We note that our results (Theorem 2, Corollary 3, Theorem 5) applies to **all interpolators** trained using **any algorithm**. Thus, even robustly trained classifiers suffer from adversarial error if they interpolate noisy labels. However, as has been recently shown in [7], AT tends to not memorise label noise. We are not aware of any work that proves why AT does not memorise label noise; and we think that would be an interesting direction of research by itself but outside the scope of this work.
>
> References.
> [1] Muthukumar, Vidya, et al. "Harmless interpolation of noisy data in regression." IEEE Journal on Selected Areas in Information Theory 1.1 (2020): 67-83.
> [2] Bartlett, Peter L., et al. "Benign overfitting in linear regression." Proceedings of the National Academy of Sciences 117.48 (2020): 30063-30070.
> [3] Wang, Ke, and Christos Thrampoulidis. "Binary classification of Gaussian mixtures: abundance of support vectors, benign overfitting and regularization." arXiv preprint arXiv:2011.09148 (2020).
> [4] Chatterji, Niladri S., and Philip M. Long. "Finite-sample Analysis of Interpolating Linear Classifiers in the Overparameterized Regime." J. Mach. Learn. Res. 22 (2021): 129-1.
> [5] Donhauser, Konstantin, et al. "Fast rates for noisy interpolation require rethinking the effects of inductive bias." arXiv preprint arXiv:2203.03597 (2022).
> [6] Wang, G., et al.  Tight bounds for minimum $\ell_1$-norm interpolation of noisy data. Proceedings of The 25th International Conference on Artificial Intelligence and Statistics, 2022.
> [7] Zhu, Jianing, et al. "Understanding the interaction of adversarial training with noisy labels." arXiv preprint arXiv:2102.03482 (2021).

---

> > ### Comment · Reviewer_59Qr · 2022-12-02
> > **Thanks**
> >
> > Thank you for the detailed response. I have no further questions and maintain my positive recommendation.

---

### Official Review · Reviewer_HmKE · 2022-10-30

**Confidence:** 3
**Correctness:** 3
**Technical Novelty And Significance:** 2
**Empirical Novelty And Significance:** 3
**Recommendation:** 6

**Clarity, Quality, Novelty And Reproducibility:**

Clarity: At a local level, the paper is quite clear and easy to follow. However, I was unfortunately struggling to get a big picture and unified view on the results. Section 3 was particularly confusing and did not add to the flow of the rest of the paper, which already felt like a combination of disjoint pieces.

(Major)
i. Could the authors please clarify why the assumption of Sanyal et al. about input distribution consisting of balls with a single class is unrealistic? It seems like an assumption on the separation between classes which seems natural.

(Minor)
(a) In the introduction, it is not clear how this work relates to Sanyal et al. What is natural vs adversarial vs uniform random label noise? It needs to be motivated
(b) the presentation didn't make it very clear that $f$ is any interpolant. The definition was buried in the setup

Quality: The theoretical results are generally precise and look accurate (I didn't verify all proofs), and the experimental conclusions are well substantiated. The paper makes a series of conjectures that seem a little unnecessary, and it was hard to get clear takeaways from those parts. The paper mentions inductive biases, but offers no real-world "obvious" experiments like comparing interpolants obtained via adversarial training to interpolants obtained via ERM.

Originality: The paper builds heavily off of the work of Sanyal et al., and extends their results to more general settings. Experiments on different kinds of label noise seem novel and interesting.

**Strength And Weaknesses:**

Strengths: The paper studies a generally important problem of understanding the robustness of interpolants (like modern deep networks) in the presence of label noise. The result on how uniform label noise is as hard as worst-case noise is surprising and interesting (but might be of limited practical use as discussed below). The paper also performs experiments on real-world label noise (dataset from prior work) and observes that the adversarial risk in the presence of real world human annotator noise is much better than under uniform label noise.

Weaknesses: The first result that generalizes Sanyal's result to more general input distributions does not seem that significant/useful to me. As noted below, the authors should clarify why their assumptions are more reflective of real-world datasets than their current assumptions. I also do not believe this paper offers any actionable insights, apart from possibly the insight that "real world" noise is easier to handle than uniform label noise. But of course, there is the difficulty of actually modeling real-world noise correctly. The paper does not compare different training methods (or different interpolants) which seems like an obvious missing piece to me.

**Summary Of The Paper:**

The paper studies the adversarial robustness of interpolants (that fit the training data perfectly) in the presence of label noise. They provide bounds on the risk of any interpolant under some mild assumptions on the data generating process. They also prove that adversarial robustness under uniform label noise is close to the worst-case label noise for fixed budget of incorrect labels. They also study the effect of different forms of label noise on real-world datasets and find that "naturally occuring" label noise leads to less adversarial susceptibility than uniform label noise.

**Summary Of The Review:**

The paper offers some interesting insights, but is missing a clear overall story or actionable takeaways. I believe the paper is currently borderline and I hope the authors can clarify some concerns raised above that can help tilt away from borderline.

---

> ### Author Response · Authors · 2022-11-13
> **Reply to Reviewer HmKE**
>
> We thank the reviewer for their very careful reading, insightful suggestions, and overall positive assessment of our work. We are happy that the reviewer found our result on the comparison between worst case, uniformly random, and human annotator noise interesting. We address the points raised by the reviewer below.
>
> **Regarding Assumption in theorem 1**
>
> We agree that the word *unrealistic* is perhaps too strong and we removed it in the text. However, we think being able to relax assumptions and prove stronger results on an important topic like this is interesting. Nevertheless, we argue below with **new experiments in Appendix F**, why we think that the assumptions are strict.
>  * The two assumptions in Theorem 1 imply that, in $\ell_{\infty}$ or $\ell_2$ distance there are regions in the input space, which contain data points of the same label (assumption 1) and these high density regions contain no points of the opposite label.
>  * A way to validate this would be to see whether there is a significant difference in the distribution of distances between points of the same label and those of the opposing label. We plot the density function of this distribution in Figure 6 in the appendix and show that the difference is indeed negligible.
>
> Here, we also want to point out that our result is indeed “stronger” than the existing result as discussed in the two paragraphs below theorem 2. Ofcourse, the importance of the result varies between readers and we note that while this reviewer found the results in section 4 to be more interesting than section 2, some of the other reviewers found the opposite.
>
> **Inductive bias but no experiments**
>
> We have made the following two changes to address this:
> * We have added new experiments in Appendix G which shows that neural networks trained using ERM on noisy data (vs noiseless data) shows the same inductive bias as the one used in our section (Theorem 7) on inductive bias.
> * We have also referred to an existing work which shows that adversarial training actually does not memorise label noise, thereby making it harder to obtain interpolators. In general, we think that understanding the optimisation dynamics (and hence the inductive bias) of neural network training algorithms like Adversarial training are indeed interesting but not yet well understood and outside the scope of the paper. Please see our general comments to see a more detailed discussion about the relevance of the inductive bias in theorem 7.
>
> **Conjectures in the paper**
>
> We have only made one conjecture in the paper, which we believe is an interesting and technically non-trivial open problem. We have also discussed why we think this is an interesting conjecture and how it relates to existing work, which another reviewer has appreciated. Apart from that, we have proven all the other theoretical statements in the paper.

---

> > ### Author Response · Authors · 2022-11-13
> > **Bigger picture and connecting the results**
> >
> > Big picture: We wanted to write a few sentences in bullet points to highlight the big picture of the paper.
> > * Previous work has shown that interpolating label noise necessarily hurts adversarial robustness. However, their result did not mention whether it could be improved. In this paper,
> >     * We use fewer assumptions than the previous paper and prove a stronger result (Theorem 2, Corollary 3).
> >     * We show our result is tight (proposition 4) and this line of work (proving lower bound on adversarial risk for **any** interpolant) cannot hope to achieve better results than ours.
> > * Next, we look at two different noise models. One where the points, whose labels are flipped, are chosen by a malicious adversary and the other where real world people label the points and unintentionally introduce noise.
> >     * We show that analysing the impact of uniform label noise (Theorem 5), as we do in this paper, is nearly sufficient  for understanding the impact of malicious adversary, which is a more different problem.
> >     * We show experimentally that real world label noise is less harmful than uniform label noise. We provide a conceptual example (in Proposition 6) why that might be the case.
> > * Finally, going back to our initial point  about  ``this line of work (proving lower bound on adversarial risk for **any** interpolant) cannot hope to achieve significantly better results’’,
> >     * We prove that if we assume that the interpolants have certain inductive biases then the results can be made much stronger (Theorem 7).
> >     * We show experimentally that neural networks indeed exhibit those kinds of inductive biases (Appendix G). This was also suggested by [2].
> >
> >
> > In short, we thank the reviewer for clearly mentioning their questions and suggestions to tilt the paper away from borderline. We are happy to engage and respond to any further questions.
> >
> > References.
> > [1] Zhu, Jianing, et al. "Understanding the interaction of adversarial training with noisy labels." arXiv preprint arXiv:2102.03482 (2021).
> > [2] Shamir, Adi, Odelia Melamed, and Oriel BenShmuel. "The dimpled manifold model of adversarial examples in machine learning." arXiv preprint arXiv:2106.10151 (2021).

---

### Official Review · Reviewer_eERJ · 2022-10-31

**Confidence:** 4
**Clarity, Quality, Novelty And Reproducibility:** See the section above.
**Correctness:** 4
**Technical Novelty And Significance:** 2
**Empirical Novelty And Significance:** Not applicable
**Recommendation:** 6

**Strength And Weaknesses:**

**Strength**:
1. **Writing**: I feel this paper is clearly written. All theorems are easy to understand, and the authors have presented proof sketches for most of them.
2. **Relevance**: The topic of adversarial robustness is clearly important to the research community, and the authors studied how adversarial vulnerability comes from as well as discussing sample complexity and inductive biases, which can be regarded as fundamental problems in this area.
3. **Significance**: The bound of Theorem 2 improves prior works, and Proposition 4 shows that it is tight.

**Weaknesses**:
1. **Limited technical novelty**. I think a major weakness is that most theoretical results in this paper are trivial to some extent. It is obvious that when label noise is present, an interpolating classifier much fit the noise and thus all the neighboring samples must be adversarially vulnerable. In particular, after reading Theorem 2 or Proposition 4, I basically have an idea of what the proofs look like. They did not surprise me or give me much new insight. While this paper indeed improves the bound of Sanyal et al. (2021) who first discovered the relationship between label noise and adversarial robustness, I think tightening the bound of Sanyal et al. (2021) is not a difficult task, so in this respect the contribution of this paper is limited. Note that Sanyal et al. (2021) already pointed out that they ``focused on making conceptually clear statements rather than to get the best possible bounds''.
2. **Incomplete theoretical results**. A number of results in this paper are either proposed as intuitions or conjectures, or are justified by giving toy examples.
- The authors demonstrate that if the data distribution is long-tailed and the label noise is more likely to occur in the long tail, then the adversarial risk is large. The intuition is quite easy to understand since a low label noise in regions with high probability mass is already enough to incur high adversarial risk in that region. However, the authors only gave a very toy example (Proposition 6) which is not really convincing to me. I may suggest the authors rigorously define the notion of long-tailed distribution and present a formal theorem.
- The authors hypothesize that real-world noise is more benign than uniform label noise. However, they did not show why this happen. Moreover, I think it *contradicts the previous argument* that label noise in the long tail is more harmful. The authors wrote that ``in the real world, examples in the long tail, are more likely to be mislabeled''. So why does the adversarial risk become lower according to Figure 3?
- To show the importance of inductive bias, the authors presented Theorem 7 which is again a toy example. I am not clear how a classifier with a T-shaped boundary relates to practical settings. On the other hand, the setting of Conjecture 1 is general, non-trivial, and important in my opinion, but the authors did not give any justification on why Conjecture 1 may hold.
3. **Inconsistency with empirical results**. As the authors said, Theorem 2 and Proposition 4 require a prohibitive sample size that does not match the empirical finding. The authors have attributed the reason to the non-uniform label noise and inductive bias. However, the authors did not further derive theoretical results in a proper setting that matches the empirical finding. It is said that ``real-world noise is more benign than uniform label noise'', yet the required sample size is much lower (e.g., on MNIST or CIFAR-10). How can this phenomenon be explained? As for the inductive bias, the author did not give satisfactory results on what inductive bias current neural networks have and how can the inductive bias hurt robustness. If the authors can prove Conjecture 1 (or a weaker version), the quality of this paper can be significantly improved.

**Summary Of The Paper:**

This paper is a follow-up work of Sanyal et al. (2021), who first discovered how label noise and inductive bias of neural networks lead to adversarial vulnerability. The contribution of this paper can be summarized as follows (where I have ordered them according to the importance in my opinion):
- Theorem 2 improves the result of Sanyal et al. (2021) in case of uniform label noise, providing a lower bound the sample size $m$ required such that with high probability the adversarial risk is $\Omega(1)$. It relaxed several assumptions in Sanyal et al. (2021) while getting a better bound on $m$.
- Proposition 4 shows that the bound of Theorem 2 is tight by giving a toy example.
- Theorem 5 shows that uniform label noise is almost as strong as poisoning, i.e., selecting an arbitrary subset of samples with a fixed size and changing labels of selected samples.
- Other contributions, including (1) an observation that label noise by human mistake is not as strong as uniform label noise; (2) a hypothesis that label noise for long-tailed samples is very harmful; (3) a toy example as well as a conjecture that the inductive bias of neural networks influences the adversarial risk.

**Summary Of The Review:**

On the one hand, the theoretical results of this paper are correct and improve the prior work of Sanyal et al. (2021). The topic of this paper as well as the writing are generally good.

On the other hand, currently the theoretical contribution seems marginal. Most of these theorems are easy to prove and does not carry much insight. A number of conclusions are made by giving toy examples or by raising as a conjecture, and some are not well-justified.

---

> ### Author Response · Authors · 2022-11-13
> **Reply to Reviewer eERJ (1/2)**
>
> We thank the reviewer for their effort: a careful reading of the paper, very valuable suggestions, and a high quality review overall. Below, we comment on the specific questions of the reviewer, and clear up the mentioned concerns:
>
> **Technicality of proof**
>
> We think it’s in fact a strong point of our paper that the exposition makes the proofs intuitive to strong researchers. It is of course hard to argue about whether a proof is technically difficult and we believe this should not be a major factor in discussing the significance of a paper.
> In addition the level of complexity and required tools vary for different results: (Theorem 2, Proposition 4, Theorem 5) including some novel constructions in Proposition 6 and Theorem 7. For other researchers working in this domain, the proofs might not be as intuitive and may inspire future work. Several reviewers (BqSH, sRSH) have also explicitly mentioned that the theoretical work is non-trivial, and others mentioned it gave them new insights.
>
> In addition, we believe and multiple reviewers (including yourself) agree with us that the results in our paper are about an important topic in machine learning. We are the first to present this **sharp bounds**  on required sample size for a lower bound on adversarial risk for interpolating classifiers. From a theoretical perspective, this presents a complete picture (both upper bound (Section 2) and matching lower bound (Section 3) ) of what can be said about adversarial risk for interpolating classifiers. **In addition, we believe theorem 5 is the first result to link the adversarial error induced by uniform label noise and that induced by a poisoning adversary.**
>
>
> **Relevance of inductive bias in Theorem 7**
>
> Thank you for this important question. We agree that this is an important point to consider;  in Appendix G, we have included **new experiments  with neural networks to show that this behavior, with T-shaped decision boundary, is a toy example similar to what is observed by neural networks**. The reason why the T-shaped decision regions in Theorem 7 amplify the effect of label noise to adversarial risk is that the decision boundaries induced by label noise are not orthogonal to the data manifold. **There is empirical evidence [1,2,3] to believe that adversarial vulnerability for neural networks are at least partly due to the same underlying reason as Theorem 7**.  While these papers do not consider label noise, they do support the “inductive bias” that leads to large adversarial error. We show this phenomenon more explicitly  by plotting the decision boundaries for neural networks that interpolate label noise in Appendix G.
>
>
> **Long tail and label noise**
>
> We think there might have been a misunderstanding about the statement of Proposition 6 here. The intuition provided by the reviewer that _“low label noise in high density region”_ leads to large adversarial error is correct.  But the statement that _“if the distribution is long-tailed and the label noise is more likely to occur in the tail then the adversarial error is large”_ is incorrect.
> Instead, as we prove in Proposition 6, label noise being concentrated in the long tail leads to smaller adversarial error.  Thus, **there is no contradiction with Figure 3 and our statement that “in the real world label noise is more likely to occur in the long tail”**.

---

> > ### Author Response · Authors · 2022-11-13
> > **Reply to Reviewer eERJ (2/2)**
> >
> > **Incomplete theoretical results**
> >
> >  Our main results are Theorem 2 (associated with Corollary 3), Proposition 3, and Theorem 5. All of these results are complete without any conjectures or intuitions. The remaining results (Proposition 6, Theorem 7, and Conjecture 1) tries to present a fuller picture of our story by addressing the limitations of the main results. To do this, we require using additional specific assumptions (Proposition 6 and Theorem 7), which we then validate using experiments.
> > * The assumption of Proposition 6 is relevant for the experiments in Figure 3
> > * The assumptions of Theorem 7 are relevant for the experiments in Appendix G.
> >
> > We hope this helps to clarify the main results, which are not incomplete and the supporting smaller results, which we support with experimental results.
> >
> >
> > **Did not derive a theoretical results that match the empirical findings, proving conjecture 1 and explicitly stating inductive bias of neural networks**
> >
> > Providing a theoretical result that matches the findings of Figure 2 or proving Conjecture 1 requires a **much better understanding of the optimization dynamics and inductive biases of neural networks**, which is an important but orthogonal field of research and is well beyond the scope of the paper. On the other hand, we are the first to show i) what is the best achievable guarantee without any assumptions on the inductive bias, ii) provide an example how the achieved guarantee can be improved significantly with an assumption on inductive bias iii) show the assumed inductive bias is similar to what is observed in practice.
> >
> > ** "real-world noise is more benign than uniform label noise", yet the required sample size is much lower (e.g., on MNIST or CIFAR-10)** This is consistent with our study. Let us explain why that is the case.
> > * We said, in Section 3, based on experimental observations in Figure 2, that “the required sample size is much lower” as an answer to the question “ for a given **uniform noise** rate, what is the amount of required data that can guarantee the lower bound on adversarial error?”. In other words, experiments show that even for a low sample size, interpolating uniform label noise induces large adversarial error.
> > * On the other hand, “real world noise is more benign that uniform label noise” simply says that for the same noise rate, the adversarial error incurred due to interpolating real world noise is much smaller than that incurred due to interpolating uniform label noise. This does not contradict the previous statement as they are both statements on different noise models.
> >
> >
> > To conclude, we would like to stress the following and we will make it clearer in the paper that we consider our main theoretical results to be captured in Theorem 2, Corollary 3, Proposition 4, and Theorem 5. Proposition 6 and Theorem 7 are additional results, where we show how certain assumptions can be used to improve the results and we provide experimental validations of those assumptions in Figure 3 and Appendix G respectively.  Please also refer to the general comments, where we summarise the bigger picture of this work.
> >
> > **We hope our answers have helped answer your questions regarding our work, and hope you will consider increasing your score. If you have any further questions or suggestions we would be more than happy to engage in answering them.**
> >
> > References
> > [1] Stutz, David, Matthias Hein, and Bernt Schiele. "Disentangling adversarial robustness and generalization." Proceedings of the IEEE/CVF Conference on Computer Vision and Pattern Recognition. 2019.
> > [2] Shamir, Adi, Odelia Melamed, and Oriel BenShmuel. "The dimpled manifold model of adversarial examples in machine learning." 2021.
> > [3] Ma, Xingjun, et al. "Characterizing adversarial subspaces using local intrinsic dimensionality."  International Conference on Learning Representations. 2018.

---

> > > ### Comment · Reviewer_eERJ · 2022-11-25
> > > **Thank you for the detailed response.**
> > >
> > > I would like to thank the authors for their detailed response as well as paper revision.
> > > - The experimental results in Appendix G seems interesting but I am still confused on why this may happen. In particular, when there is label noise, the T-shaped boundary appears; but when there is no label noise, the boundary is quite normal. It seems that the inductive bias is insufficient to explain this. Could the authors give some intuitive explanations?
> > > - Regarding "long tail and label noise". My concern has been suitably addressed.
> > > - Regarding real-world noise and sample size. The theory is definitely correct due to the assumption of uniform noise, but it does not align well with practice. So we may reach the agreement that the assumption used in this paper is unrealistic in practice.
> > > - As for the technical novelty or the completeness of the results, while I am still not convinced by the author response, I think it may be hard to reach an agreement.
> > >
> > > Acknowledgement: I have read the general response and the revised paper.
> > >
> > > Some minor comments:
> > > - In page 22, Figure 8, the caption says: "The first row in each box is without label noise, and the second row is with label noise." Is it written wrong?
> > > - Typo:  In Proposition 6:  "Let the ground truth label be zero everwhere". It should be "everywhere".
> > >
> > > Overall, I tend towards slightly increasing the score to 6 considering the authors' efforts in answering so many questions raised by different reviewers.

---

> > > > ### Author Response · Authors · 2022-12-06
> > > > **Response to Reviewer eERj**
> > > >
> > > > Dear Reviewer,
> > > >
> > > > Thank you for your reply, for engaging in the discussion, and for tending towards increasing the score. We reply to your comments below
> > > >
> > > > * Regarding long-tail and label noise, we are happy that we have been able to answer your concern.
> > > >
> > > > * Regarding real world noise and sample size, we agree that theorem 2,3 is impractical and we argue the same in the paper. So, as the reviewer says, we believe we are in agreement here. This is the reason why we also discuss non-uniform noise and inductive bias.
> > > >
> > > > * Regarding inductive bias: Inductive bias of algorithms is a very broad term that not only dictates how the clean label is fit but also how noisy labels are fit. The two ways in which these two tasks are accomplished can be very different as highlighted in our experiments. This is also the focus of many works, that we have referred to in our rebuttal,  that try to explain how noisy labels are interpolated in neural networks in simpler theoretical settings. While explaining this is out of scope for us, we highlight some hypotheses here.
> > > >
> > > >     One hypothesis for why the T shaped boundary occurs for fitting noisy points is the combination of the following two intuitive properties
> > > >     1. The decision region of the neural networks has low sensitivity (thus smoother) on the training data manifold (XY axis) but is very sensitive (and can thus be non-smooth) off the training data manifold (Z axis). Thus, if required the non-smoothness occurs along the z axis and not on the x-y plane.
> > > >      2. Preference towards large connected regions in the decision regions of the neural network.
> > > >
> > > > Both points have been argued previously in many papers. See section 4.2 in Novak et. al. for arguments on the first point and see Fawzi et. al. for arguments on the second point. If both of these points are satisfied for clean data, then you would not expect to see any T shaped region as they can be satisfied by the smooth linear boundary, which is invariant to the z dimension.
> > > > However, fitting noisy data requires “smaller” decision regions on the XY axis around the noisy points. Any such region is part of a  larger decision region of the same label that is either above it or below it on the z axis. Due to the possibly non-smoothness along the z-axis, this larger region can be very close to the data manifold, as shown in our experiments.
> > > >
> > > > To conclude, we again want to highlight that we do not claim this explanation to be a part of our paper, but more in the spirit of discussion with the reviewer regarding possible answers to their interesting questions.
> > > >
> > > > We also wanted to lightly mention to the reviewer that the reviewer had mentioned  that they are tending to increase the score to 6 but we can see the score is still 5.
> > > >
> > > > Novak, Roman, et al. "Sensitivity and generalization in neural networks: an empirical study." arXiv preprint arXiv:1802.08760 (2018).
> > > > Fawzi, Alhussein, et al. "Classification regions of deep neural networks." arXiv preprint arXiv:1705.09552 (2017).

---

> > > > > ### Comment · Reviewer_eERJ · 2022-12-07
> > > > > **Thank you for answering my remaining questions.**
> > > > >
> > > > > I am satisfied with the answers about the T-shaped boundary here. I think the explanation is interesting and you may make some discussions in the Appendix in the camera-ready version. Also, the reason I haven't raised my score is because I'm waiting for your reply. (I don't think you need to rush me on this before answering the remaining questions).

---

### Official Review · Reviewer_sRSH · 2022-11-02

**Confidence:** 3
**Correctness:** 4
**Technical Novelty And Significance:** 3
**Empirical Novelty And Significance:** 2
**Recommendation:** 6

**Clarity, Quality, Novelty And Reproducibility:**

Clarity wise is good. While this paper focuses of theoretical justification, it is still easy for experienced reader to follow since the author provides proof sketch to every theorem and proposition. Meanwhile the discussion with respect to related work makes the contribution more solid. Toy examples also provide convincing insight. Some improvement can be done. Please see Strength And Weaknesses.
Originality and quality wise are also good. Although this paper borrows some idea on toy examples, the main theorem include non-trivial technique in its analysis. To my knowledge, these results are both novel and insightful in the subfield of adversarial robustness.

**Strength And Weaknesses:**

Strength:
1. This paper provides stronger guarantee in the same track of work by Sanyal et al. (2021) with weaker assumption. Even though the sample size in the new bound still has a heavy dependence on the covering number but science is incremental.
2. This paper provides several insightful statements which can benefit future research. Eq (8) and Example highlight potential improvement direction. Conjecture 1 also does the same job with enough literature support.

Weaknesses:
1. Some toy examples are unrealistic. Firstly, while Proposition 6 argues that some label noise models are benign, I don't believe one can choose what kind of label noise he/she wants as in the toy example.  Secondly, Theorem 7 is based on a constructed hypothesis class with a sequence of T-shaped decision regions, which is hard to image similar shape being produced by real-world dataset and deep networks but the author claims it as "an illustrative example for what might be happening in practice". I am also not clear how this setting is related to inductive bias in neural networks.

**Summary Of The Paper:**

This paper studies the adversarial risk caused by label noise. In particular, Theorem 2 provides a constant lower bound on the adversarial risk of an interpolator in the presence of uniform label noise when the sample size is large enough. Such bound relaxes the assumption and improves over required sample size in the previous results (Sanyal et al., 2021). Eq. (7) and Figure 2 explain Theorem 3 has no control over adversarial risk. Proposition 4 shows that the bound of Theorem 2 is sharp if arbitrary classifiers and distributions are allowed. Theorem 5 considers the game theory perspective and shows if the uniform label noise is given double the adversarial radius and a log factor increase on the training set size, then the risk is as strong as data poisoning. Proposition 6 provides two toy examples of long-tailed data distribution and shows that the interpolating can be benign. Figure 3 shows that uniform label noise is worse for adversarial risk than human-generated label noise. Theorem 7 shows the inductive bias of the function class makes the impact of label noise on adversarial vulnerability much stronger.

**Summary Of The Review:**

This paper improves state-of-art theoretical guarantees and it can be an important advance in the sub-field. Although the theoretical finding does not align with empirical phenomenon perfectly, I still believe the solid analysis and insight outweigh the limitation.

---

> ### Author Response · Authors · 2022-11-13
> **Reply to reviewer  sRSH**
>
> We thank the reviewer for their careful reading, helpful comments, very accurate summary, and overall positive assessment of our work.
>
> We are also happy that our effort into providing proof sketches was appreciated by the reviewer. In addition to appreciating the reviewer’s comments about the bound in theorem 2 being interesting despite being incremental, we also want to point out that proposition 4 essentially says that the bound cannot be improved any further. We believe, combined, these two results have a much larger significance than them individually as they present a complete picture of the problem.
>
> **Regarding label noise model**
>
> We agree that we the learner cannot choose the model of label noise present in the data and thus, similar to other problems in the computer security literature, any guarantee should reflect the worst possible noise model i.e. a poisoning adversary. This is precisely what we do in Theorem 5, where we show that the effect of uniform label noise is almost as bad as the effect of an adversary who can perturb any data points’ label as they choose as long as the total number of such points are the same.
>
> The “benign” label noise model is an empirical result (Figure 3(a), 3(b)) where we show that label noise in human annotations, obtained from a publicly available dataset (http://www.noisylabels.com/) has a lesser impact than uniform label noise (which is nearly as bad as the poisoning model). In Proposition 6, combined with Figure 3(c), we intended to provide an explanation for why that might be the case. So while Proposition 6 uses a constructed distribution, we believe that, combined with Figure 3(c), it provides an explanation for the effect observed in real world experiments Figure 3(a,b).
>
> **Regarding T-shaped decision regions in theorem 7**
>
> We have now included an appendix (Appendix G) where we provide experimental results to connect theorem 7 and its illustration in Figure 4 to the actual decision region of realistic experiments using neural networks. We hope this helps clarify the reference to inductive bias. However, we would also like to note that characterising inductive bias is a longstanding problem without a precise definition. Our theorem 7 provides an example and Appendix G shows that the example in theorem 7 is not very far from what is observed in neural networks. We included a slightly more detailed discussion in the general comments.
>
>
> To conclude, we appreciate that the reviewer found several insights in our results including in the toy examples. We hope our general comments and the comments above help convince the reviewer that the toy examples are indeed, at least partially, relevant for observations in real  world experiments. We hope the reviewer will consider improving their assessment of our work as well. We are happy to answer any more questions if the reviewer has any.

---

### Official Review · Reviewer_BXAC · 2022-11-04

**Confidence:** 3
**Correctness:** 4
**Technical Novelty And Significance:** 3
**Empirical Novelty And Significance:** Not applicable
**Recommendation:** 6

**Clarity, Quality, Novelty And Reproducibility:**

The paper is written clear and has improved the previous work with a tighter provable bound.

**Strength And Weaknesses:**

Strength:
* The paper is written well with clear structure.
* The theoretical result improves previous works with a tighter training sample size lower bound.

Weaknesses:
* The tightness of the bound is mostly dependent on the choice of subset C, which is unknown how to get an optimized one.
* The theorem can not explain the adversarial risk of the model when trained with  clean data where $\eta = 0$, which requires infinite sample size for a given adversarial risk lower bound.

**Summary Of The Paper:**

This paper studies the connection between label noise and adversarial risk by given a theorem on the sample size bound for a given adversarial risk with certain noise rate. This theoretical result improves previous work on sample size lower bound. The author also proves that their theorem is tight under a constructed special circumstance.

**Summary Of The Review:**

I think this paper is a good paper to analyze the adversarial risk resulted from label noise and it has a tighter result compared to previous works. But the main theorem is still not enough to explain the cause of adversarial risk completely due to the difficulty in finding optimal C and also its infinite bound when $\eta = 0$.

---

> ### Author Response · Authors · 2022-11-13
> **Official reply to Reviewer BXAC**
>
> We thank the reviewer for their careful reading of the paper, their comments, and their overall positive feedback of our work! We address the specific comments of the reviewer below.
>
> **Main theorem is still not enough to explain the cause of adversarial risk completely ... its infinite bound when $\eta = 0$.**
>
> It is true that our results do not speak for the case of $\eta = 0$, because that value of $\eta~(=0)$ corresponds to only clean labels in the dataset. We focused our work, specifically, on the  contribution of label noise to adversarial risk. Thus, while it is true that there might be other possible causes of adversarial risk (for example when there is probability mass of the data close to the ground truth decision boundary), our result says how much label noise contributes to adversaries risk even when there are no other causes of adversarial risk and a robust interpolator is achievable in the noiseless setting. This does not contradict the $\eta=0$ case, as it is possible that there are distributions where, if the dataset is noiseless, a perfect robust classifier can be learned.
>
> **Main theorem is still not enough to explain the cause of adversarial risk completely ... due to the difficulty in finding optimal C**
>
> Regarding the choice of $C$, we view this as a strength of our result and we can hopefully convince the reviewer to view it as so. First, corollary 2 essentially gets rid of $C$ and is also tight as pointed out by Proposition 2. However, it has a larger sample size requirement and guarantees large adversarial risk. On the other hand, being able to choose $C$, allows us to adapt to different distributions and sample sizes and give tighter results for that case. We provide two examples 1) in the third last paragraph of Page 5 and 2) 2nd part of the proof of Proposition 6 and the first part of the proof of Theorem 7 also relies on this. In addition, we would like to highlight that solving an optimisation problem within a bound to get a tighter bound is not unique to our work and has been used before eg. in SQ dimension for SQ learnability[1], representation dimension for  private learnability[2], and even confidence sequences [3].
>
>
> In addition, we’d like to highlight that we have a **second “main” result**, Theorem 5 about how uniform label noise is surprisingly close to worst-case label poisoning. We hope that this answers all the questions of the reviewer and hope that  the reviewer will consider increasing their score for our work. We are also happy to engage further should the reviewer have any more questions.
>
> References.
> [1]  Feldman, Vitaly. "A general characterization of the statistical query complexity." Conference on Learning Theory. PMLR, 2017.
> [2] Beimel, Amos, Kobbi Nissim, and Uri Stemmer. "Characterizing the sample complexity of private learners." Proceedings of the 4th conference on Innovations in Theoretical Computer Science. 2013.
> [2] Waudby-Smith, Ian, and Aaditya Ramdas. "Estimating means of bounded random variables by betting." arXiv preprint arXiv:2010.09686 (2020).

---

### Author Response · Authors · 2022-11-13
**General comments to everyone**

General comments

We thank all the reviewers for the positive and helpful comments. We’re glad that we got  a large number of informative reviews and that the summaries of the reviews are generally agreeing with the picture that we have of our work.
* We are glad that almost all reviewers agree that Theorem 2 is an improvement over previous work, while also being elegant (**BqSH,  59QR**) and using non-trivial technique (**sRSH**).
* Several reviewers (**HmKE,59Qr**) highlighted that Theorem 5, is surprising and interesting in its own right. Recall that there we show that uniform label noise is almost as harmful as worst-case label noise.
* Some reviewers (**UsFF,AQu9**) lauded the experiment in Figure 3 and appreciated the use of the idea of memorization score. To recall, this experiment experimentally supports the theoretical intuition about uniform label noise being worse than real world label noise in human annotations.
* Finally, almost all reviewers think that the paper is well-written and easy to follow, and that our proof sketches are clear.


For convenience, here we summarize and address the main concerns raised by reviewers, which we think would be useful to everyone. Our updated in the main text are marked in blue:


**A big picture to relate the different contributions**
* Previous work has shown that interpolating label noise necessarily hurts adversarial robustness. However, their result did not mention whether it could be improved. In this paper,
    * We use _fewer assumptions_ than the previous paper and prove a stronger result (Theorem 2, Corollary 3).
    * We show our result is tight (proposition 4) and this line of work (proving lower bound on adversarial risk for **any** interpolant) cannot hope to achieve better results than ours.
* Next, we look at _two different noise models_. One where the points, whose labels are flipped, are chosen by a malicious adversary and the other where real world people label the points and unintentionally introduce noise.
    * We show that analysing the impact of uniform label noise (Theorem 5), as we do in this paper, is nearly sufficient  for understanding the impact of malicious adversary, which is a more different problem.
    * We show experimentally that real world label noise is less harmful than uniform label noise. We provide a conceptual example (in Proposition 6) why that might be the case.
* Finally, going back to our initial point  about  _``this line of work (proving lower bound on adversarial risk for **any interpolant**) cannot hope to achieve significantly better results’’_,
    * We prove that if we assume that the interpolants have certain inductive biases then the results can be made much stronger (Theorem 7).
    * We show experimentally that small neural networks indeed exhibit those kinds of inductive biases (Appendix G). This was also suggested by [2].

---

> ### Author Response · Authors · 2022-11-13
> **General replies to some questions**
>
> **Inductive bias in Theorem 7 and in Neural networks**
>
> Some reviewers (sRSH, eERJ, BqSH) rightly questioned whether the hypothesis class in Theorem 7 indeed reflects the inductive bias of neural networks. The theorem was intended as an example of why inductive bias is an important part of this discussion and how it can **significantly decrease** the adversarial error. We provide **new experiments in Appendix G**, where we argue why it is relevant for neural networks.
>
> In light of the first two subsections in Section 3, there must be some property of the decision of neural networks that amplifies the adversarial risk in neural networks, compared to arbitrary interpolating classifiers. In theorem 7, the reason why the T-shaped decision regions amplify the effect of label noise to adversarial risk is that the decision boundaries induced by label noise are orthogonal to the data manifold. There is empirical evidence [1,2,3] to believe that adversarial vulnerability for neural networks is at least partly due to the same underlying reason as Theorem 7.  While these papers do not consider label noise, they do support the “inductive bias” that leads to large adversarial error. We show this phenomenon more specifically by plotting the decision boundaries for neural networks that interpolate label noise in Appendix G.
>
> **Assumptions in theorem 1**
>
> A reviewer (HmKE) asked about what we gain from eliminating the assumption on the input distribution consisting of high density balls each containing a single class. We argue below with **new experiments in Appendix F**, why we think that the assumptions are strict.  However, we do agree that we should not use the word "unreasonable" but rather "strict"
>
> The two assumptions in Theorem 1 imply that, in $\ell_{\infty}$ or $\ell_2$ distance there are regions in the input space, which contain data points of the same label (assumption 1) and these high density regions contain no points of the opposite label.
> A way to validate this would be to see whether there is a significant difference in the distribution of distances between points of the same label and those of opposing label. We plot the density function of this distribution in Figure 6 in the appendix and show that the difference is indeed negligible.
>
> In addition to having less assumptions, we also want to point out that our result is indeed “stronger” than Theorem 1 in some cases when the assumptions of both are satisfied.. We discuss this in the second paragraph below Theorem 2, but to recap:
> Imagine a distribution containing $N$ disjoint balls, and draw $m = \Omega(\frac{N}{\eta c})$ samples, for some constant $c$. Then Theorem 1 requires each of the balls to have probability mass on the order of $c$, while Theorem 2 only requires the average ball density to be on the order of $c$.
>
> Finally, several reviewers agree that Theorem 2 is more elegant and interpretable than Theorem 1, especially in the context of the last paragraph of Section 3.
>
> References:
>  [1] Stutz, David, Matthias Hein, and Bernt Schiele. "Disentangling adversarial robustness and generalization." Proceedings of the IEEE/CVF Conference on Computer Vision and Pattern Recognition. 2019.
> [2] Shamir, Adi, Odelia Melamed, and Oriel BenShmuel. "The dimpled manifold model of adversarial examples in machine learning." 2021.
> [3] Ma, Xingjun, et al. "Characterizing adversarial subspaces using local intrinsic dimensionality."  International Conference on Learning Representations. 2018.

---

### Decision · Program_Chairs · 2023-01-20

**Decision:**

Accept: poster

**Justification For Why Not Higher Score:**

- As the authors honestly admit, there are limitations of the authors' theorems and they explore potential ways to improve them to bring them closer to practice.

**Justification For Why Not Lower Score:**

- The paper provides several interesting insights, for example,
  -  human-generated label noise leads to less adversarial susceptibility than uniform label noise.
  - uniform label noise can be nearly as harmful as worst-case data poisoning
- The main result (Theorem 2) improves upon previous work and is shown to be tight

**Metareview: Summary, Strengths And Weaknesses:**

Summary
---
This paper studies the relationship between label noise and adversarial risk. The authors prove a theorem showing that interpolating label noise can lead to adversarial vulnerability, and demonstrate that this result is almost tight when no assumptions are made on the inductive bias of the learning algorithm. They also investigate how different factors, including the properties of the data distribution, impact this result. They show that uniform label noise can lead to nearly as large an adversarial risk as worst data poisoning, and provide both theoretical and empirical evidence that real-world label noise is less harmful than uniform label noise. Finally, they demonstrate that inductive biases can amplify the effect of label noise.

Strengths
---
- Theorem 2 improves upon previous work and has a tighter training sample size lower bound.
- It shows that uniform label noise can be nearly as harmful as worst case data poisoning, given a slight increase in dataset size and adversarial radius. It also performs experiments on real-world label noise and observes that human-generated label noise leads to less adversarial susceptibility than uniform label noise.
- It provides insights on the impact of different forms of label noise and inductive bias on adversarial vulnerability.
- It is well-written and presents clear intuition for its theoretical results. It also explores potential ways to improve the results to bring them closer to practical use.

Weakness
---
- The tightness of the bound for Theorem 2 may not be practical for standard datasets. The theorem also does not provide a bound for the adversarial risk of a model trained with clean data ($\eta=0$), which requires an infinite sample size for a given adversarial risk lower bound. The authors acknowledged this point in their response.
- As the authors honestly admit, there are limitations of the authors' theorems and they explore potential ways to improve them to bring them closer to practice.



**Note From Pc:**

if the above contains the word "oral" or "spotlight" please see: "oral" presentation means -> notable-top-5% and "spotlight" means -> notable-top-25%. As stated in our emails, we are disassociating presentation type from AC recommendations